# EPISODE: Episodic Gradient Clipping with Periodic Resampled Corrections for Federated Learning with Heterogeneous Data

**Michael Crawshaw**[1], **Yajie Bao**[2], **Mingrui Liu**[1]*
[1]Department of Computer Science, George Mason University, Fairfax, VA 22030, USA
[2]School of Mathematical Sciences, Shanghai Jiao Tong University, Shanghai, China
`mcrawsha@gmu.edu, baoyajie2019stat@sjtu.edu.cn, mingruil@gmu.edu`

## Abstract

Gradient clipping is an important technique for deep neural networks with exploding gradients, such as recurrent neural networks. Recent studies have shown that the loss functions of these networks do not satisfy the conventional smoothness condition, but instead satisfy a relaxed smoothness condition, i.e., the Lipschitz constant of the gradient scales linearly in terms of the gradient norm. Due to this observation, several gradient clipping algorithms have been developed for nonconvex and relaxed-smooth functions. However, the existing algorithms only apply to the single-machine or multiple-machine setting with homogeneous data across machines. It remains unclear how to design provably efficient gradient clipping algorithms in the general Federated Learning (FL) setting with heterogeneous data and limited communication rounds. In this paper, we design EPISODE, the very first algorithm to solve FL problems with heterogeneous data in the nonconvex and relaxed smoothness setting. The key ingredients of the algorithm are two new techniques called *episodic gradient clipping* and *periodic resampled corrections*. At the beginning of each round, EPISODE resamples stochastic gradients from each client and obtains the global averaged gradient, which is used to (1) determine whether to apply gradient clipping for the entire round and (2) construct local gradient corrections for each client. Notably, our algorithm and analysis provide a unified framework for both homogeneous and heterogeneous data under any noise level of the stochastic gradient, and it achieves state-of-the-art complexity results. In particular, we prove that EPISODE can achieve linear speedup in the number of machines, and it requires significantly fewer communication rounds. Experiments on several heterogeneous datasets, including text classification and image classification, show the superior performance of EPISODE over several strong baselines in FL. The code is available at `https://github.com/MingruiLiu-ML-Lab/episode`.

## 1 Introduction

Gradient clipping (Pascanu et al., 2012; 2013) is a well-known strategy to improve the training of deep neural networks with the exploding gradient issue such as Recurrent Neural Networks (RNN) (Rumelhart et al., 1986; Elman, 1990; Werbos, 1988) and Long Short-Term Memory (LSTM) (Hochreiter & Schmidhuber, 1997). Although it is a widely-used strategy, formally analyzing gradient clipping in deep neural networks under the framework of nonconvex optimization only happened recently (Zhang et al., 2019a; 2020a; Cutkosky & Mehta, 2021; Liu et al., 2022). In particular, Zhang et al. (2019a) showed empirically that the gradient Lipschitz constant scales linearly in terms of the gradient norm when training certain neural networks such as AWD-LSTM (Merity et al., 2018), introduced the relaxed smoothness condition (i.e., $(L_0, L_1)$-smoothness[1]), and proved that clipped gradient descent converges faster than any fixed step size gradient descent. Later on, Zhang et al. (2020a) provided tighter complexity bounds of the gradient clipping algorithm.

Federated Learning (FL) (McMahan et al., 2017a) is an important distributed learning paradigm in which a single model is trained collaboratively under the coordination of a central server without revealing client data [2]. FL has two critical features: heterogeneous data and limited communication.

---

*Corresponding Author: Mingrui Liu (`mingruil@gmu.edu`).
[1]The formal definition of $(L_0, L_1)$-smoothness is illustrated in Definition 2.
[2]In this paper, we use the terms "client" and "machine" interchangeably.

Table 1: Communication complexity ($R$) and largest number of skipped communication ($I_{\max}$) to guarantee linear speedup for different methods to find an $\epsilon$-stationary point (defined in Definition 1). "Single" means single machine, $N$ is the number of clients, $I$ is the number of skipped communications, $\kappa$ is the quantity representing the heterogeneity, $\Delta = f(\boldsymbol{x}_0) - \min_{\boldsymbol{x}} f(\boldsymbol{x})$, and $\sigma^2$ is the variance of stochastic gradients. Iteration complexity ($T$) is the product of communication complexity and the number of skipped communications (i.e., $T = RI$ ). Best iteration complexity $T_{\min}$ denotes the minimum value of $T$ the algorithm can achieve through adjusting $I$. Linear speedup means the iteration complexity is divided by $N$ compared with the single machine baseline: in our case it means $T = \mathcal{O}(\frac{\Delta L_0\sigma^2}{N\epsilon^4})$ iteration complexity.

| Method | Setting | Communication Complexity ($R$) | Best Iteration Complexity ($T_{\min}$) | Largest $I$ to guarantee linear speedup ($I_{\max}$) |
|---|---|---|---|---|
| Local SGD (Yu et al., 2019c) | Heterogeneous, $L$-smooth | $\mathcal{O}\left(\frac{\Delta L\sigma^2}{NI\epsilon^4} + \frac{\Delta L\kappa^2 NI}{\sigma^2\epsilon^2} + \frac{\Delta LN}{\epsilon^2}\right)$ | $\mathcal{O}(\frac{\Delta L\sigma^2}{N\epsilon^4})$ | $\mathcal{O}\left(\frac{\sigma^2}{\kappa N\epsilon}\right)$ |
| SCAFFOLD (Karimireddy et al., 2020) | Heterogeneous, $L$-smooth | $\mathcal{O}\left(\frac{\Delta L\sigma^2}{NI\epsilon^4} + \frac{\Delta L}{\epsilon^2}\right)$ | $\mathcal{O}(\frac{\Delta L\sigma^2}{N\epsilon^4})$ | $\mathcal{O}\left(\frac{\sigma^2}{N\epsilon^2}\right)$ |
| Clipped SGD (Zhang et al., 2019b) | Single, $(L_0, L_1)$-smooth | $\mathcal{O}\left(\frac{\left(\Delta+(L_0+L_1\sigma)\sigma^2+\sigma L_0^2/L_1\right)^2}{\epsilon^4}\right)$ | $\mathcal{O}\left(\frac{\left(\Delta+(L_0+L_1\sigma)\sigma^2+\sigma L_0^2/L_1\right)^2}{\epsilon^4}\right)$ | N/A |
| Clipping Framework (Zhang et al., 2020a) | Single, $(L_0, L_1)$-smooth | $\mathcal{O}\left(\frac{\Delta L_0\sigma^2}{\epsilon^4}\right)$ | $\mathcal{O}\left(\frac{\Delta L_0\sigma^2}{\epsilon^4}\right)$ | N/A |
| CELGC (Liu et al., 2022) | Homogeneous, $(L_0, L_1)$-smooth | $\mathcal{O}\left(\frac{\Delta L_0\sigma^2}{NI\epsilon^4}\right)$ | $\mathcal{O}(\frac{\Delta L_0\sigma^2}{N\epsilon^4})$ | $\mathcal{O}\left(\frac{\sigma}{N\epsilon}\right)$ |
| EPISODE (this work) | Heterogeneous, $(L_0, L_1)$-smooth | $\mathcal{O}\left(\frac{\Delta L_0\sigma^2}{NI\epsilon^4} + \frac{\Delta(L_0+L_1(\kappa+\sigma))}{\epsilon^2}\left(1+\frac{\sigma}{\epsilon}\right)\right)$ | $\mathcal{O}(\frac{\Delta L_0\sigma^2}{N\epsilon^4})$ | $\mathcal{O}\left(\frac{L_0\sigma^2}{(L_0+L_1(\kappa+\sigma))(1+\frac{\sigma}{\epsilon})N\epsilon^2}\right)$ |

Although there is a vast literature on FL (see (Kairouz et al., 2019) and references therein), the theoretical and algorithmic understanding of gradient clipping algorithms for training deep neural networks in the FL setting remains nascent. To the best of our knowledge, Liu et al. (2022) is the only work that has considered a communication-efficient distributed gradient clipping algorithm under the nonconvex and relaxed smoothness conditions in the FL setting. In particular, Liu et al. (2022) proved that their algorithm achieves linear speedup in terms of the number of clients and reduced communication rounds. Nevertheless, their algorithm and analysis are only applicable to the case of homogeneous data. In addition, the analyses of the stochastic gradient clipping algorithms in both single machine (Zhang et al., 2020a) and multiple-machine setting (Liu et al., 2022) require strong distributional assumptions on the stochastic gradient noise [3], which may not hold in practice.

In this work, we introduce a provably computation and communication efficient gradient clipping algorithm for nonconvex and relaxed-smooth functions in **the general FL setting (i.e., heterogeneous data, limited communication)** and **without any distributional assumptions on the stochastic gradient noise**. Compared with previous work on gradient clipping (Zhang et al., 2019a; 2020a; Cutkosky & Mehta, 2020; Liu et al., 2022) and FL with heterogeneous data (Li et al., 2020a; Karimireddy et al., 2020), our algorithm design relies on two novel techniques: *episodic gradient clipping* and *periodic resampled corrections*. In a nutshell, at the beginning of each communication round, the algorithm resamples each client's stochastic gradient; this information is used to decide whether to apply clipping in the current round (i.e., *episodic gradient clipping*), and to perform local corrections to each client's update (i.e., *periodic resampled corrections*). These techniques are very different compared with previous work on gradient clipping. Specifically, (1) In traditional gradient clipping (Pascanu et al., 2012; Zhang et al., 2019a; 2020a; Liu et al., 2022), whether or not to apply the clipping operation is determined only by the norm of the client's current stochastic gradient. Instead, we use the norm of the global objective's stochastic gradient (resampled at the beginning of the round) to determine whether or not clipping will be applied throughout the entire communication round. (2) Different from Karimireddy et al. (2020) which uses historical gradient information from the previous round to perform corrections, our algorithm utilizes the resampled gradient to correct each client's local update towards the global gradient, which mitigates the effect of data heterogeneity. Notice that, under the relaxed smoothness setting, the gradient may change quickly around a point at which the gradient norm is large. Therefore, our algorithm treats a small gradient as more "reliable" and confidently applies the unclipped corrected local updates; on the contrary, the algorithm regards a large gradient as less "reliable" and in this case clips the corrected local updates. Our major contributions are summarized as follows.

---

[3] Zhang et al. (2020a) requires an explicit lower bound for the stochastic gradient noise, and Liu et al. (2022) requires the distribution of the stochastic gradient noise is unimodal and symmetric around its mean.

- We introduce EPISODE, the very first algorithm for optimizing nonconvex and $(L_0, L_1)$-smooth functions in the general FL setting with heterogeneous data and limited communication. The algorithm design introduces novel techniques, including episodic gradient clipping and periodic resampled corrections. To the best of our knowledge, these techniques are first introduced by us and crucial for algorithm design.

- Under the nonconvex and relaxed smoothness condition, we prove that the EPISODE algorithm can achieve linear speedup in the number of clients and reduced communication rounds in the heterogeneous data setting, without any distributional assumptions on the stochastic gradient noise. In addition, we show that the degenerate case of EPISODE matches state-of-the-art complexity results under weaker assumptions [4]. Detailed complexity results and a comparison with other relevant algorithms are shown in Table 1.

- We conduct experiments on several heterogeneous medium and large scale datasets with different deep neural network architectures, including a synthetic objective, text classification, and image classification. We show that the performance of the EPISODE algorithm is consistent with our theory, and it consistently outperforms several strong baselines in FL.

## 2 RELATED WORK

**Gradient Clipping** Gradient clipping is a standard technique in the optimization literature for solving convex/quasiconvex problems (Ermoliev, 1988; Nesterov, 1984; Shor, 2012; Hazan et al., 2015; Mai & Johansson, 2021; Gorbunov et al., 2020), nonconvex smooth problems (Levy, 2016; Cutkosky & Mehta, 2021), and nonconvex distributionally robust optimization (Jin et al., 2021). Menon et al. (2019) showed that gradient clipping can help mitigate label noise. Gradient clipping is a well-known strategy to achieve differential privacy (Abadi et al., 2016; McMahan et al., 2017b; Andrew et al., 2021; Zhang et al., 2021). In the deep learning literature, gradient clipping is employed to avoid exploding gradient issue when training certain deep neural networks such as recurrent neural networks or long-short term memory networks (Pascanu et al., 2012; 2013) and language models (Gehring et al., 2017; Peters et al., 2018; Merity et al., 2018). Zhang et al. (2019a) initiated the study of gradient clipping for nonconvex and relaxed smooth functions. Zhang et al. (2020a) provided an improved analysis over Zhang et al. (2019a). However, none of these works apply to the general FL setting with nonconvex and relaxed smooth functions.

**Federated Learning** FL was proposed by McMahan et al. (2017a), to enable large-scale distributed learning while keep client data decentralized to protect user privacy. McMahan et al. (2017a) designed the FedAvg algorithm, which allows multiple steps of gradient updates before communication. This algorithm is also known as local SGD (Stich, 2018; Lin et al., 2018; Wang & Joshi, 2018; Yu et al., 2019c). The local SGD algorithm and their variants have been analyzed in the convex setting (Stich, 2018; Stich et al., 2018; Dieuleveut & Patel, 2019; Khaled et al., 2020; Li et al., 2020a; Karimireddy et al., 2020; Woodworth et al., 2020a;b; Koloskova et al., 2020; Yuan et al., 2021) and nonconvex smooth setting (Jiang & Agrawal, 2018; Wang & Joshi, 2018; Lin et al., 2018; Basu et al., 2019; Haddadpour et al., 2019; Yu et al., 2019c;b; Li et al., 2020a; Karimireddy et al., 2020; Reddi et al., 2021; Zhang et al., 2020b; Koloskova et al., 2020). Recently, in the stochastic convex optimization setting, several works compared local SGD and minibatch SGD in the homogeneous (Woodworth et al., 2020b) and heterogeneous data setting (Woodworth et al., 2020a), as well as the fundamental limit (Woodworth et al., 2021). For a comprehensive survey, we refer the readers to Kairouz et al. (2019); Li et al. (2020a) and references therein. The most relevant work to ours is Liu et al. (2022), which introduced a communication-efficient distributed gradient clipping algorithm for nonconvex and relaxed smooth functions. However, their algorithm and analysis does not apply in the case of heterogeneous data as considered in this paper.

## 3 PROBLEM SETUP AND PRELIMINARIES

**Notations** In this paper, we use $\langle \cdot, \cdot \rangle$ and $\| \cdot \|$ to denote the inner product and Euclidean norm in space $\mathbb{R}^d$. We use $\mathbb{1}(\cdot)$ to denote the indicator function. We let $\mathcal{I}_r$ be the set of iterations at the $r$-th

---

[4]We prove that the degenerate case of our analysis (e.g., homogeneous data) achieves the same iteration and communication complexity, but without the requirement of unimodal and symmetric stochastic gradient noise as in Liu et al. (2022). Also, our analysis is unified over any noise level of stochastic gradient, which does not require an explicit lower bound for the stochastic gradient noise as in the analysis of Zhang et al. (2020a).

round, that is $\mathcal{I}_r = \{t_r, ..., t_{r+1} - 1\}$. The filtration generated by the random variables before step $t_r$ is denoted by $\mathcal{F}_r$. We also use $\mathbb{E}_r[\cdot]$ to denote the conditional expectation $\mathbb{E}[\cdot|\mathcal{F}_r]$. The number of clients is denoted by $N$ and the length of the communication interval is denoted by $I$, i.e., $|\mathcal{I}_r| = I$ for $r = 0, 1, ..., R$. Let $f_i(\boldsymbol{x}) := \mathbb{E}_{\xi_i \sim \mathcal{D}_i}[F_i(\boldsymbol{x}; \xi_i)]$ be the loss function in $i$-th client for $i \in [N]$, where the local distribution $\mathcal{D}_i$ is unknown and may be different across $i \in [N]$. In the FL setting, we aim to minimize the following overall averaged loss function:

$$f(\boldsymbol{x}) := \frac{1}{N} \sum_{i=1}^{N} f_i(\boldsymbol{x}). \tag{1}$$

We focus on the case that each $f_i$ is non-convex, in which it is NP-hard to find the global minimum of $f$. Instead we consider finding an $\epsilon$-stationary point (Ghadimi & Lan, 2013; Zhang et al., 2020a).

**Definition 1.** *For a function $h : \mathbb{R}^d \to \mathbb{R}$, a point $x \in \mathbb{R}^d$ is called $\epsilon$-stationary if $\|\nabla h(\boldsymbol{x})\| \le \epsilon$.*

Most existing works in the non-convex FL literature (Yu et al., 2019a; Karimireddy et al., 2020) assume each $f_i$ is $L$-smooth, i.e., $\|\nabla f_i(\boldsymbol{x}) - \nabla f_i(\boldsymbol{y})\| \le L\|\boldsymbol{x} - \boldsymbol{y}\|$ for any $\boldsymbol{x}, \boldsymbol{y} \in \mathbb{R}^d$. However it is shown in Zhang et al. (2019a) that $L$-smoothness may not hold for certain neural networks such as RNNs and LSTMs. $(L_0, L_1)$-smoothness in Definition 2 was proposed by Zhang et al. (2019b) and is strictly weaker than $L$-smoothness. Under this condition, the local smoothness of the objective can grow with the gradient scale. For AWD-LSTM (Merity et al., 2018), empirical evidence of $(L_0, L_1)$-smoothness was observed in Zhang et al. (2019b).

**Definition 2.** *A second order differentiable function $h : \mathbb{R}^d \to \mathbb{R}$ is $(L_0, L_1)$-smooth if $\|\nabla^2 h(\boldsymbol{x})\| \le L_0 + L_1\|\nabla h(\boldsymbol{x})\|$ holds for any $x \in \mathbb{R}^d$.*

Suppose we only have access to the stochastic gradient $\nabla F_i(\boldsymbol{x}; \xi)$ for $\xi \sim \mathcal{D}_i$ in each client. Next we make the following assumptions on objectives and stochastic gradients.

**Assumption 1.** *Assume $f_i$ for $i \in [N]$ and $f$ defined in (1) satisfy:*

(i) *$f_i$ is second order differentiable and $(L_0, L_1)$-smooth.*

(ii) *Let $\boldsymbol{x}^*$ be the global minimum of $f$ and $\boldsymbol{x}_0$ be the initial point. There exists some $\Delta > 0$ such that $f(\boldsymbol{x}_0) - f(\boldsymbol{x}^*) \le \Delta$.*

(iii) *For all $\boldsymbol{x} \in \mathbb{R}^d$, there exists some $\sigma \ge 0$ such that $\mathbb{E}_{\xi_i \sim \mathcal{D}_i}[\nabla F_i(\boldsymbol{x}; \xi_i)] = \nabla f_i(\boldsymbol{x})$ and $\|\nabla F_i(\boldsymbol{x}; \xi_i) - \nabla f_i(\boldsymbol{x})\| \le \sigma$ almost surely.*

(iv) *There exists some $\kappa \ge 0$ and $\rho \ge 1$ such that $\|\nabla f_i(\boldsymbol{x})\| \le \kappa + \rho\|\nabla f(\boldsymbol{x})\|$ for any $\boldsymbol{x} \in \mathbb{R}^d$.*

**Remark**: (i) and (ii) are standard in the non-convex optimization literature (Ghadimi & Lan, 2013), and (iii) is a standard assumption in the $(L_0, L_1)$-smoothness setting (Zhang et al., 2019b; 2020a; Liu et al., 2022). (iv) is used to bound the difference between the gradient of each client's local loss and the gradient of the overall loss, which is commonly assumed in the FL literature with heterogeneous data (Karimireddy et al., 2020). When $\kappa = 0$ and $\rho = 1$, (iv) corresponds to the homogeneous setting.

## 4 ALGORITHM AND ANALYSIS

### 4.1 MAIN CHALLENGES AND ALGORITHM DESIGN

**Main Challenges** We first illustrate why the prior local gradient clipping algorithm (Liu et al., 2022) would not work in the heterogeneous data setting. Liu et al. (2022) proposed the first communication-efficient local gradient clipping algorithm (CELGC) in a homogeneous setting for relaxed smooth functions, which can be interpreted as the clipping version of FedAvg. Let us consider a simple heterogeneous example with two clients in which CELGC fails. Denote $f_1(x) = \frac{1}{2}x^2 + a_1 x$ and $f_2(x) = \frac{1}{2}x^2 + a_2 x$ with $a_1 = -\gamma - 1$, $a_2 = \gamma + 2$, and $\gamma > 1$. We know that the optimal solution for $f = \frac{f_1 + f_2}{2}$ is $x_* = -\frac{a_1 + a_2}{2} = -\frac{1}{2}$, and both $f_1$ and $f_2$ are $(L_0, L_1)$-smooth with $L_0 = 1$ and $L_1 = 0$. Consider CELGC with communication interval $I = 1$ (i.e., communication happens at every iteration), starting point $x_0 = 0$, $\eta = 1/L_0 = 1$, clipping threshold $\gamma$, and $\sigma = 0$. In this setting, after the first iteration, the model parameters on the two clients become $\gamma$ and $-\gamma$ respectively, so that the averaged model parameter after communication returns to $0$. This means that the model parameter of CELGC remains stuck at $0$ indefinitely, demonstrating that CELGC cannot handle data heterogeneity.

---

**Algorithm 1:** Episodic Gradient Clipping with Periodic Resampled Corrections (EPISODE)

---

1: Initialize $\boldsymbol{x}_0^i \leftarrow \boldsymbol{x}_0, \bar{\boldsymbol{x}}_0 \leftarrow \boldsymbol{x}_0$.
2: **for** $r = 0, 1, ..., R$ **do**
3:    **for** $i \in [N]$ **do**
4:       Sample $\nabla F_i(\bar{\boldsymbol{x}}_r; \widetilde{\xi}_r^i)$ where $\widetilde{\xi}_r^i \sim \mathcal{D}_i$, and update $\boldsymbol{G}_r^i \leftarrow \nabla F_i(\bar{\boldsymbol{x}}_r; \widetilde{\xi}_r^i)$.
5:    **end for**
6:    Update $\boldsymbol{G}_r = \frac{1}{N} \sum_{i=1}^N \boldsymbol{G}_r^i$.
7:    **for** $i \in [N]$ **do**
8:       **for** $t = t_r, \ldots, t_{r+1} - 1$ **do**
9:          Sample $\nabla F_i(\boldsymbol{x}_t^i; \xi_t^i)$, where $\xi_t^i \sim \mathcal{D}_i$, and compute $\boldsymbol{g}_t^i \leftarrow \nabla F_i(\boldsymbol{x}_t^i; \xi_t^i) - \boldsymbol{G}_r^i + \boldsymbol{G}_r$.
10:          $\boldsymbol{x}_{t+1}^i \leftarrow \boldsymbol{x}_t^i - \eta \boldsymbol{g}_t^i \mathbb{1}(\|\boldsymbol{G}_r\| \leq \gamma/\eta) - \gamma \frac{\boldsymbol{g}_t^i}{\|\boldsymbol{g}_t^i\|} \mathbb{1}(\|\boldsymbol{G}_r\| \geq \gamma/\eta)$.
11:       **end for**
12:    **end for**
13:    Update $\bar{\boldsymbol{x}}_r \leftarrow \frac{1}{N} \sum_{i=1}^N \boldsymbol{x}_{t_{r+1}}^i$.
14: **end for**

---

We then explain why the stochastic controlled averaging method (SCAFFOLD) (Karimireddy et al., 2020) for heterogeneous data does not work in the relaxed smoothness setting. SCAFFOLD utilizes the client gradients from the previous round to constructing correction terms which are added to each client's local update. Crucially, SCAFFOLD requires that the gradient is Lipschitz so that gradients from the previous round are good approximations of gradients in the current round with controllable errors. This technique is not applicable in the relaxed smoothness setting: the gradient may change dramatically, so historical gradients from the previous round are not good approximations of the current gradients anymore due to potential unbounded errors.

**Algorithm Design** To address the challenges brought by heterogeneity and relaxed smoothness, our idea is to clip the local updates similarly as we would clip the global gradient (if we could access it). The detailed description of EPISODE is stated in Algorithm 1. Specifically, we introduce two novel techniques: (1) Episodic gradient clipping. At the $r$-th round, EPISODE constructs a global indicator $\mathbb{1}(\|\boldsymbol{G}_r\| \leq \gamma/\eta)$ to determine whether to perform clipping in every local update during the round for all clients (line 6). (2) Periodic resampled corrections. EPISODE resamples fresh gradients with *constant batch size* at the beginning of each round (line 3-5). In particular, at the beginning of the $r$-th round, EPISODE samples stochastic gradients evaluated at the current averaged global weight $\bar{\boldsymbol{x}}_r$ in all clients to construct the control variate $\boldsymbol{G}_r$, which has two roles. The first is to construct the global clipping indicator according to $\|\boldsymbol{G}_r\|$ (line 10). The second one is to correct the bias between local gradient and global gradient through the variate $\boldsymbol{g}_t^i$ in local updates (line 10).

## 4.2 MAIN RESULTS

**Theorem 1.** *Suppose Assumption 1 holds. For any tolerance $\epsilon \leq \frac{3AL_0}{5BL_1\rho}$, we choose the hyper parameters as $\eta \leq \min\left\{\frac{1}{216\Gamma I}, \frac{\epsilon}{180\Gamma I\sigma}, \frac{N\epsilon^2}{16AL_0\sigma^2}\right\}$ and $\gamma = \left(11\sigma + \frac{AL_0}{BL_1\rho}\right)\eta$, where $\Gamma = AL_0 + BL_1\kappa + BL_1\rho\left(\sigma + \frac{\gamma}{\eta}\right)$. Then EPISODE satisfies $\frac{1}{R+1}\sum_{r=0}^R \mathbb{E}\left[\|\nabla f(\bar{\boldsymbol{x}}_r)\|\right] \leq 3\epsilon$ as long as the number of communication rounds satisfies $R \geq \frac{4\Delta}{\epsilon^2 \eta I}$.*

**Remark 1:** The result in Theorem 1 holds for arbitrary noise level, while the complexity bounds in the stochastic case of Zhang et al. (2020a); Liu et al. (2022) both require $\sigma \geq 1$. In addition, this theorem can automatically recover the complexity results in Liu et al. (2022), but does not require their symmetric and unimodal noise assumption. The improvement upon previous work comes from a better algorithm design, as well as a more careful analysis in the smoothness and individual discrepancy in the non-clipped case (see Lemma 2 and 3).

**Remark 2:** In Theorem 1, when we choose $\eta = \min\left\{\frac{1}{216\Gamma I}, \frac{\epsilon}{180\Gamma I\sigma}, \frac{N\epsilon^2}{16AL_0\sigma^2}\right\}$, the total communication complexity to find an $\epsilon$-stationary point is no more than $R = \mathcal{O}\left(\frac{\Delta}{\epsilon^2 \eta I}\right) = \mathcal{O}\left(\frac{\Delta(L_0 + L_1(\kappa + \sigma))}{\epsilon^2}\left(1 + \frac{\sigma}{\epsilon}\right) + \frac{\Delta L_0 \sigma^2}{NI\epsilon^4}\right)$. Next we present some implications of the communication complexity.

1. When $I \lesssim \frac{L_0\sigma}{(L_0+L_1(\kappa+\sigma))N\epsilon}$ and $\sigma \gtrsim \epsilon$, EPISODE has communication complexity $\mathcal{O}(\frac{\Delta L_0\sigma^2}{NI\epsilon^4})$. In this case, EPISODE enjoys a better communication complexity than the naive parallel version of the algorithm in Zhang et al. (2020a), that is $\mathcal{O}(\frac{\Delta L_0\sigma^2}{N\epsilon^4})$. Moreover, the iteration complexity of EPISODE is $T = RI = \mathcal{O}(\frac{\Delta L_0\sigma^2}{N\epsilon^4})$, which achieves the *linear speedup* w.r.t. the number of clients $N$. This matches the result of Liu et al. (2022) in the homogeneous data setting.

2. When $I \gtrsim \frac{L_0\sigma}{(L_0+L_1(\kappa+\sigma))N\epsilon}$ and $\sigma \gtrsim \epsilon$, the communication complexity of EPISODE is $\mathcal{O}(\frac{\Delta(L_0+L_1(\kappa+\sigma))\sigma}{\epsilon^3})$. This term does not appear in Theorem III of Karimireddy et al. (2020), but it appears here due to the difference in the construction of the control variates. In fact, the communication complexity of EPISODE is still lower than the naive parallel version of Zhang et al. (2020a) if the number of clients satisfies $N \lesssim \frac{L_0\sigma}{(L_0+L_1(\kappa+\sigma))\epsilon}$.

3. When $0 < \sigma \lesssim \epsilon$, EPISODE has communication complexity $\mathcal{O}(\frac{\Delta(L_0+L_1(\kappa+\sigma))}{\epsilon^2})$. Under this particular noise level, the algorithms in Zhang et al. (2020a); Liu et al. (2022) do not guarantee convergence because their analyses crucially rely on the fact that $\sigma \gtrsim \epsilon$.

4. When $\sigma = 0$, EPISODE has communication complexity $\mathcal{O}(\frac{\Delta(L_0+L_1\kappa)}{\epsilon^2})$. This bound includes an additional constant $L_1\kappa$ compared with the complexity results in the deterministic case (Zhang et al., 2020a), which comes from data heterogeneity and infrequent communication.

## 4.3 PROOF SKETCH OF THEOREM 1

Despite recent work on gradient clipping in the homogeneous setting (Liu et al., 2022), the analysis of Theorem 1 is highly nontrivial since we need to cope with $(L_0, L_1)$-smoothness and heterogeneity simultaneously. In addition, we do not require a lower bound of $\sigma$ and allow for arbitrary $\sigma \geq 0$.

The first step is to establish the descent inequality of the global loss function. According to the $(L_0, L_1)$-smoothness condition, if $\|\bar{\boldsymbol{x}}_{r+1} - \bar{\boldsymbol{x}}_r\| \leq C/L_1$, then

$$\mathbb{E}_r\left[f(\bar{\boldsymbol{x}}_{r+1}) - f(\bar{\boldsymbol{x}}_r)\right] \leq \mathbb{E}_r\left[(\mathbb{1}(\mathcal{A}_r) + \mathbb{1}(\bar{\mathcal{A}}_r))\langle\nabla f(\bar{\boldsymbol{x}}_r), \bar{\boldsymbol{x}}_{r+1} - \bar{\boldsymbol{x}}_r\rangle\right]$$
$$+ \mathbb{E}_r\left[(\mathbb{1}(\mathcal{A}_r) + \mathbb{1}(\bar{\mathcal{A}}_r))\frac{AL_0 + BL_1\|\nabla f(\bar{\boldsymbol{x}}_r)\|}{2}\|\bar{\boldsymbol{x}}_{r+1} - \bar{\boldsymbol{x}}_r\|^2\right], \quad (2)$$

where $\mathcal{A}_r := \{\|\boldsymbol{G}_r\| \leq \gamma/\eta\}$, $\bar{\mathcal{A}}_r$ is the complement of $\mathcal{A}_r$, and $A, B, C$ are constants defined in Lemma 5. To utilize the inequality (2), we need to verify that the distance between $\bar{\boldsymbol{x}}_{r+1}$ and $\bar{\boldsymbol{x}}_r$ is small almost surely. In the algorithm of Liu et al. (2022), clipping is performed in each iteration based on the magnitude of the current stochastic gradient, and hence the increment of each local weight is bounded by the clipping threshold $\gamma$. For each client in EPISODE, whether to perform clipping is decided by the magnitude of $\boldsymbol{G}_r$ at the beginning of each round. Therefore, the techniques in Liu et al. (2022) to bound the individual discrepancy cannot be applied to EPISODE. To address this issue, we introduce Lemma 1, which guarantees that we can apply the properties of relaxed smoothness (Lemma 5 and 6) to all iterations in one round, in either case of clipping or non-clipping.

**Lemma 1.** *Suppose* $2\eta I(AL_0 + BL_1\kappa + BL_1\rho(\sigma + \gamma/\eta)) \leq 1$ *and* $\max\{2\eta I(2\sigma + \gamma/\eta), \gamma I\} \leq \frac{C}{L_1}$. *Then for any* $i \in [N]$ *and* $t - 1 \in \mathcal{I}_r$, *it almost surely holds that*

$$\mathbb{1}(\mathcal{A}_r)\|\boldsymbol{x}_t^i - \bar{\boldsymbol{x}}_r\| \leq 2\eta I(2\sigma + \gamma/\eta) \quad and \quad \mathbb{1}(\bar{\mathcal{A}}_r)\|\boldsymbol{x}_t^i - \bar{\boldsymbol{x}}_r\| \leq \gamma I. \quad (3)$$

Equipped with Lemma 1, the condition $\|\bar{\boldsymbol{x}}_{r+1} - \bar{\boldsymbol{x}}_r\| \leq \frac{1}{N}\sum_{i=1}^{N}\|\boldsymbol{x}_{t_{r+1}}^i - \bar{\boldsymbol{x}}_r\| \leq C/L_1$ can hold almost surely with a proper choice of $\eta$. Then it suffices to bound the terms from (2) in expectation under the events $\mathcal{A}_r$ and $\bar{\mathcal{A}}_r$ respectively. To deal with the discrepancy term $\mathbb{E}[\|\boldsymbol{x}_t^i - \bar{\boldsymbol{x}}_r\|^2]$ for $t - 1 \in \mathcal{I}_r$, Liu et al. (2022) directly uses the almost sure bound for both cases of clipping and non-clipping. Here we aim to obtain a more delicate bound in expectation for the non-clipping case. The following lemma, which is critical to obtain the unified bound from Theorem 1 under any noise level, gives an upper bound for the local smoothness of $f_i$ at $\boldsymbol{x}$.

**Lemma 2.** *Under the conditions of Lemma 1, for all* $\boldsymbol{x} \in \mathbb{R}^d$ *such that* $\|\boldsymbol{x} - \bar{\boldsymbol{x}}_r\| \leq 2\eta I(2\sigma + \gamma/\eta)$, *the following inequality almost surely holds:*

$$\mathbb{1}(\mathcal{A}_r)\|\nabla^2 f_i(\boldsymbol{x})\| \leq L_0 + L_1(\kappa + (\rho + 1)(\gamma/\eta + 2\sigma)).$$

From (3), we can see that all iterations in the $r$-th round satisfy the condition of Lemma 2 almost surely. Hence we are guaranteed that each local loss $f_i$ is $L$-smooth over the iterations in this round under the event $\mathcal{A}_r$, where $L = L_0 + L_1(\kappa + (\rho + 1)(\gamma/\eta + 2\sigma))$. In light of this, the following lemma gives a bound in expectation of the individual discrepancy. We denote $p_r = \mathbb{E}_r[\mathbb{1}(\mathcal{A}_r)]$.

**Lemma 3.** *Under the conditions of Lemma 1, for any $i \in [N]$ and $t - 1 \in \mathcal{I}_r$, we have*

$$\mathbb{E}_r\left[\mathbb{1}(\mathcal{A}_r)\|\boldsymbol{x}_t^i - \bar{\boldsymbol{x}}_r\|^2\right] \leq 36 p_r I^2 \eta^2 \|\nabla f(\bar{\boldsymbol{x}}_r)\|^2 + 126 p_r I^2 \eta^2 \sigma^2, \tag{4}$$

$$\mathbb{E}_r\left[\mathbb{1}(\mathcal{A}_r)\|\boldsymbol{x}_t^i - \bar{\boldsymbol{x}}_r\|^2\right] \leq 18 p_r I^2 \eta\gamma \|\nabla f(\bar{\boldsymbol{x}}_r)\| + 18 p_r I^2 \eta^2 \left(\gamma\sigma/\eta + 5\sigma^2\right). \tag{5}$$

It is worthwhile noting that the bound in (4) involves a quadratic term of $\|\nabla f(\bar{\boldsymbol{x}}_r)\|$, whereas it is linear in (5). The role of the linear bound is to deal with $\mathbb{1}(\mathcal{A}_r)\|\nabla f(\bar{\boldsymbol{x}}_r)\|\|\bar{\boldsymbol{x}}_{r+1} - \bar{\boldsymbol{x}}_r\|^2$ from the descent inequality (2), since directly substituting (4) would result in a cubic term which is hard to analyze. With Lemma 1, 2 and 3, we obtain the following descent inequality.

**Lemma 4.** *Under the conditions of Lemma 1, let $\Gamma = AL_0 + BL_1(\kappa + \rho(\gamma/\eta + \sigma))$. Then it holds that for each $0 \leq r \leq R - 1$,*

$$\mathbb{E}_r\left[f(\bar{\boldsymbol{x}}_{r+1}) - f(\bar{\boldsymbol{x}}_r)\right] \leq \mathbb{E}_r\left[\mathbb{1}(\mathcal{A}_r)V(\bar{\boldsymbol{x}}_r)\right] + \mathbb{E}_r\left[\mathbb{1}(\bar{\mathcal{A}}_r)U(\bar{\boldsymbol{x}}_r)\right], \tag{6}$$

*where the definitions of $V(\bar{\boldsymbol{x}}_r)$ and $U(\bar{\boldsymbol{x}}_r)$ are given in Appendix C.1.*

The detailed proof of Lemma 4 is deferred in Appendix C.1. With this Lemma, the descent inequality is divided into $V(\bar{\boldsymbol{x}}_r)$ (objective value decrease during the non-clipping rounds) and $U(\bar{\boldsymbol{x}}_r)$ (objective value decrease during the clipping rounds). Plugging in the choices of $\eta$ and $\gamma$ yields

$$\max\left\{U(\bar{\boldsymbol{x}}_r), V(\bar{\boldsymbol{x}}_r)\right\} \leq -\frac{1}{4}\epsilon\eta I\|\nabla f(\bar{\boldsymbol{x}}_r)\| + \frac{1}{2}\epsilon^2 \eta I. \tag{7}$$

The conclusion of Theorem 1 can then be obtained by substituting (7) into (6) and summing over $r$.

## 5 EXPERIMENTS

In this section, we present an empirical evaluation of EPISODE to validate our theory. We present results in the heterogeneous FL setting on three diverse tasks: a synthetic optimization problem satisfying $(L_0, L_1)$-smoothness, natural language inference on the SNLI dataset (Bowman et al., 2015), and ImageNet classification (Deng et al., 2009). We compare EPISODE against FedAvg (McMahan et al., 2017a), SCAFFOLD (Karimireddy et al., 2020), CELGC (Liu et al., 2022), and a naive distributed algorithm which we refer to as Naive Parallel Clip [5] We include additional experiments on the CIFAR-10 dataset (Krizhevsky et al., 2009) in Appendix E.4, running time results in Appendix F, ablation study in Appendix G, and new experiments on federated learning benchmark datasets in Appendix H.

### 5.1 SETUP

All non-synthetic experiments were implemented with PyTorch (Paszke et al., 2019) and run on a cluster with eight NVIDIA Tesla V100 GPUs. Since SNLI , CIFAR-10, and ImageNet are centralized datasets, we follow the non-i.i.d. partitioning protocol in (Karimireddy et al., 2020) to split each dataset into heterogeneous client datasets with varying label distributions. Specifically, for a similarity parameter $s \in [0, 100]$, each client's local dataset is composed of two parts. The first $s\%$ is comprised of i.i.d. samples from the complete dataset, and the remaining $(100 - s)\%$ of data is sorted by label.

**Synthetic** To demonstrate the behavior of EPISODE and baselines under $(L_0, L_1)$-smoothness, we consider a simple minimization problem in a single variable. Here we have $N = 2$ clients with:

$$f_1(x) = x^4 - 3x^3 + Hx^2 + x, \quad f_2(x) = x^4 - 3x^3 - 2Hx^2 + x,$$

where the parameter $H$ controls the heterogeneity between the two clients. Notice that $f_1$ and $f_2$ satisfy $(L_0, L_1)$-smoothness but not traditional $L$-smoothness.

**Proposition 1.** *For any $x \in \mathbb{R}$ and $i = 1, 2$, it holds that $\|\nabla f_i(x)\| \leq 2\|\nabla f(x)\| + \kappa(H)$, where $\kappa(H) < \infty$ and is a positive increasing function of $H$ for $H \geq 1$.*

According to Proposition 1, Assumption 1(iv) will be satisfied with $\rho = 2$ and $\kappa = \kappa(H)$, where $\kappa(H)$ is an increasing function of $H$. The proof of this proposition is deferred to Appendix E.1.

---

[5] Naive Parallel Clip uses the globally averaged stochastic gradient obtained from synchronization at every iteration to run SGD with gradient clipping on the global objective.

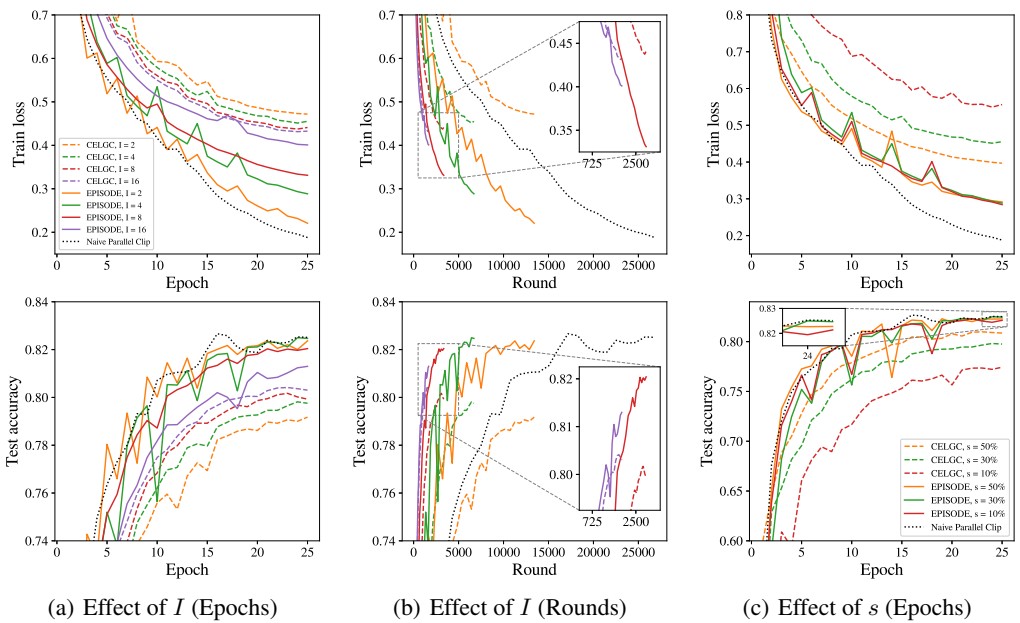

(a) Effect of $I$ (Epochs)    (b) Effect of $I$ (Rounds)    (c) Effect of $s$ (Epochs)

Figure 1: Training loss and testing accuracy on SNLI. The style of each curve (solid, dashed, dotted) corresponds to the algorithm, while the color corresponds to either the communication interval $I$ (for (a) and (b)) or the client data similarity $s$ (for (c)). **(a), (b)** Effect of varying $I$ with $s = 30\%$, plotted against (a) epochs and (b) communication rounds. **(c)** Effect of varying $s$ with $I = 4$.

**SNLI** Following Conneau et al. (2017), we train a BiRNN network for 25 epochs using the multi-class hinge loss and a batch size of 64 on each worker. The network is composed of a one layer BiRNN encoder with hidden size 2048 and max pooling, and a three-layer fully connected classifier with hidden size 512. The BiRNN encodes a sentence (represented as a sequence of GloVe vectors (Pennington et al., 2014)), and the classifier predicts the relationship of two encoded sentences as either entailment, neutral, or contradiction. For more hyperparameter information, see Appendix E.3.

To determine the effects of infrequent communication and data heterogeneity on the performance of each algorithm, we vary $I \in \{2, 4, 8, 16\}$ and $s \in \{10\%, 30\%, 50\%\}$. We compare EPISODE, CELGC, and the Naive Parallel Clip. Note that the training process diverged when using SCAFFOLD, likely due to a gradient explosion issue, since SCAFFOLD does not use gradient clipping.

**ImageNet** Following Goyal et al. (2017), we train a ResNet-50 (He et al., 2016) for 90 epochs using the cross-entropy loss, a batch size of 32 for each worker, clipping parameter $\gamma = 1.0$, momentum with coefficient 0.9, and weight decay with coefficient $5 \times 10^{-5}$. We initially set the learning rate $\eta = 0.1$ and decay by a factor of 0.1 at epochs 30, 60, and 80. To analyze the effect of data heterogeneity in this setting, we fix $I = 64$ and vary $s \in \{50\%, 60\%, 70\%\}$. Similarly, to analyze the effect of infrequent communication, we fix $s = 60\%$ and vary $I \in \{64, 128\}$. We compare the performance of FedAvg, CELGC, EPISODE, and SCAFFOLD.

## 5.2 RESULTS

**Synthetic** Figure 3 in Appendix E.2 shows the objective value throughout training, where the heterogeneity parameter $H$ varies over $\{1, 2, 4, 8\}$. CELGC exhibits very slow optimization due to the heterogeneity across clients: as $H$ increases, the optimization progress becomes slower and slower. In contrast, EPISODE maintains fast convergence as $H$ varies. We can also see that EPISODE converges to the minimum of global loss, while CELGC fails to do so when $H$ is larger.

**SNLI** Results for the SNLI dataset are shown in Figure 1. To demonstrate the effect of infrequent communication, Figures 1(a) and 1(b) show results for EPISODE, CELGC, and Naive Parallel Clip as the communication interval $I$ varies (with fixed $s = 30\%$). After 25 epochs, the test accuracy

| Interval | Similarity | Algorithm | Train loss | Test acc. |
|----------|-----------|-----------|------------|-----------|
| 64 | 70% | FedAvg | 1.010 | 74.89% |
|    |     | CELGC | 1.016 | 74.89% |
|    |     | SCAFFOLD | 1.024 | 74.92% |
|    |     | EPISODE | **0.964** | **75.20%** |
| 64 | 60% | FedAvg | 0.990 | 74.73% |
|    |     | CELGC | 0.979 | 74.51% |
|    |     | SCAFFOLD | 0.983 | 74.68% |
|    |     | EPISODE | **0.945** | **74.95%** |
| 64 | 50% | FedAvg | 0.955 | 74.53% |
|    |     | CELGC | 0.951 | 74.12% |
|    |     | SCAFFOLD | 0.959 | 74.19% |
|    |     | EPISODE | **0.916** | **74.81%** |
| 128 | 60% | FedAvg | 1.071 | 74.15% |
|     |     | CELGC | 1.034 | 74.24% |
|     |     | SCAFFOLD | 1.071 | 74.03% |
|     |     | EPISODE | **1.016** | **74.36%** |

Figure 2: ImageNet results. **Left:** Training loss and testing accuracy at the end of training for various settings of $I$ and $s$. EPISODE consistently reaches better final metrics in all settings. **Right:** Training loss and testing accuracy during training for $I = 64$ and $s = 50\%$.

of EPISODE nearly matches that of Naive Parallel Clip for all $I \leq 8$, while CELGC lags 2-3% behind Naive Parallel Clip for all values of $I$. Also, EPISODE nearly matches the test accuracy of Naive Parallel Clip with as little as $8$ times fewer communication rounds. Lastly, EPISODE requires significantly less communication rounds to reach the same training loss as CELGC. For example, EPISODE with $I = 4$, $s = 30\%$ takes less than $5000$ rounds to reach a training loss of $0.4$, while CELGC does not reach $0.4$ during the entirety of training with any $I$.

To demonstrate the effect of client data heterogeneity, Figure 1(c) shows results for varying values of $s$ (with fixed $I = 4$). Here we can see that EPISODE is resilient against data heterogeneity: even with client similarity as low as $s = 10\%$, the performance of EPISODE is the same as $s = 50\%$. Also, the testing accuracy of EPISODE with $s = 10\%$ is nearly identical to that of the Naive Parallel Clip. On the other hand, the performance of CELGC drastically worsens with more heterogeneity: even with $s = 50\%$, the training loss of CELGC is significantly worse than EPISODE with $s = 10\%$.

**ImageNet** Figure 2 shows the performance of each algorithm at the end of training for all settings (left) and during training for the setting $I = 64$ and $s = 50\%$ (right). Training curves for the rest of the settings are given in Appendix E.5. EPISODE outperforms all baselines in every experimental setting, especially in the case of high data heterogeneity. EPISODE is particularly dominant over other methods in terms of the training loss during the whole training process, which is consistent with our theory. Also, EPISODE exhibits more resilience to data heterogeneity than CELGC and SCAFFOLD: as the client data similarity deceases from $70\%$ to $50\%$, the test accuracies of CELGC and SCAFFOLD decrease by $0.8\%$ and $0.7\%$, respectively, while the test accuracy of EPISODE decreases by $0.4\%$. Lastly, as communication becomes more infrequent (i.e., the communication interval $I$ increases from $64$ to $128$), the performance of EPISODE remains superior to the baselines.

## 6 CONCLUSION

We have presented EPISODE, a new communication-efficient distributed gradient clipping algorithm for federated learning with heterogeneous data in the nonconvex and relaxed smoothness setting. We have proved convergence results under any noise level of the stochastic gradient. In particular, we have established linear speedup results as well as reduced communication complexity. Further, our experiments on both synthetic and real-world data show demonstrate the superior performance of EPISODE compared to competitive baselines in FL. Our algorithm is suitable for the cross-silo federated learning setting such as in healthcare and financial domains (Kairouz et al., 2019), and we plan to consider cross-device setting in the future.

ACKNOWLEDGEMENTS

We would like to thank the anonymous reviewers for their helpful comments. Michael Crawshaw is supported by the Institute for Digital Innovation fellowship from George Mason University. Michael Crawshaw and Mingrui Liu are both supported by a grant from George Mason University. The work of Yajie Bao was done when he was virtually visiting Mingrui Liu's research group in the Department of Computer Science at George Mason University.

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

## A  PRELIMINARIES

We use $\mathcal{F}_r$ to denote the filtration generated by

$$\{\xi_t^i : t \in \mathcal{I}_l, i = 1, ...N\}_{l=1}^{r-1} \cup \{\widetilde{\xi}_l^i : i = 1, ...N\}_{l=1}^{r-1}.$$

It means that given $\mathcal{F}_r$, the global solution $\bar{\boldsymbol{x}}_r$ is fixed, but the randomness of $\mathcal{A}_r$, $\boldsymbol{G}_r^i$ and $\boldsymbol{G}_r$ still exists. In addition, for $t \in \mathcal{I}_r$, we use $\mathcal{H}_t$ to denote the filtration generated by

$$\mathcal{F}_r \cup \{\xi_s^i : t_r \leq s \leq t\}_{i=1}^N \cup \{\widetilde{\xi}_r^i\}_{i=1}^N.$$

Recall the definitions of $\boldsymbol{G}_r^i$ and $\boldsymbol{G}_r$,

$$\boldsymbol{G}_r^i = \nabla F_i(\bar{\boldsymbol{x}}_r; \widetilde{\xi}_r^i) \quad \text{and} \quad \boldsymbol{G}_r = \frac{1}{N}\sum_{i=1}^N \boldsymbol{G}_r^i.$$

Hence we have

$$\|\boldsymbol{G}_r^i - \nabla f_i(\bar{\boldsymbol{x}}_r)\| \leq \sigma \quad \text{and} \quad \|\boldsymbol{G}_r - \nabla f(\bar{\boldsymbol{x}}_r)\| \leq \sigma,$$

hold almost surely due to Assumption 1(iii). Also, the local update rule of EPISODE is

$$\boldsymbol{x}_{t+1}^i = \boldsymbol{x}_t^i - \eta \boldsymbol{g}_t^i \mathbb{1}(\mathcal{A}_r) - \gamma \frac{\boldsymbol{g}_t^i}{\|\boldsymbol{g}_t^i\|}\mathbb{1}(\bar{\mathcal{A}}_r) \quad \text{for} \quad t \in \mathcal{I}_r,$$

where $\boldsymbol{g}_t^i = \nabla F_i(\boldsymbol{x}_t^i; \xi_t^i) - \boldsymbol{G}_r^i + \boldsymbol{G}_r$, $\mathcal{A}_r = \{\|\boldsymbol{G}_r\| \leq \gamma/\eta\}$ and $\bar{\mathcal{A}}_r = \{\|\boldsymbol{G}_r\| > \gamma/\eta\}$.

### A.1  AUXILIARY LEMMAS

**Lemma 5** (Lemma A.2 in Zhang et al. (2020a)). *Let $f$ be $(L_0, L_1)$-smooth, and $C > 0$ be a constant. For any $\boldsymbol{x}, \boldsymbol{x}' \in \mathbb{R}^d$ such that $\|\boldsymbol{x} - \boldsymbol{x}'\| \leq C/L_1$, we have*

$$f(\boldsymbol{x}') - f(\boldsymbol{x}) \leq \langle \nabla f(\boldsymbol{x}), \boldsymbol{x}' - \boldsymbol{x}\rangle + \frac{AL_0 + BL_1\|\nabla f(\boldsymbol{x})\|}{2}\|\boldsymbol{x}' - \boldsymbol{x}\|^2,$$

*where $A = 1 + e^C - \frac{e^C - 1}{C}$ and $B = \frac{e^C - 1}{C}$.*

**Lemma 6** (Lemma A.3 in Zhang et al. (2020a)). *Let $f$ be $(L_0, L_1)$-smooth, and $C > 0$ be a constant. For any $\boldsymbol{x}, \boldsymbol{x}' \in \mathbb{R}^d$ such that $\|\boldsymbol{x} - \boldsymbol{x}'\| \leq C/L_1$, we have*

$$\|\nabla f(\boldsymbol{x}') - \nabla f(\boldsymbol{x})\| \leq (AL_0 + BL_1\|\nabla f(\boldsymbol{x})\|)\|\boldsymbol{x}' - \boldsymbol{x}\|,$$

*where $A = 1 + e^C - \frac{e^C - 1}{C}$ and $B = \frac{e^C - 1}{C}$.*

Here we choose $C \geq 1$ such that $A \geq 1$ and $B \geq 1$.

**Lemma 7** (Lemma B.1 in Zhang et al. (2020a)). *Let $\mu > 0$ and $\boldsymbol{u}, \boldsymbol{v} \in \mathbb{R}^d$. Then*

$$-\frac{\langle \boldsymbol{u}, \boldsymbol{v}\rangle}{\|\boldsymbol{v}\|} \leq -\mu\|\boldsymbol{u}\| - (1 - \mu)\|\boldsymbol{v}\| + (1 + \mu)\|\boldsymbol{v} - \boldsymbol{u}\|.$$

## B  PROOF OF LEMMAS IN SECTION 4.3

### B.1  PROOF OF LEMMA 1

**Lemma 1 restated.**  Suppose $2\eta I(AL_0 + BL_1\kappa + BL_1\rho(\sigma + \frac{\gamma}{\eta})) \leq 1$ and $\max\left\{2\eta I(2\sigma + \frac{\gamma}{\eta}), \gamma I\right\} \leq \frac{C}{L_1}$, where the relation between $A$, $B$ and $C$ is stated in Lemma 5 and 6. Then for any $i \in [N]$ and $t - 1 \in \mathcal{I}_r$, it almost surely holds that

$$\mathbb{1}(\mathcal{A}_r)\|\boldsymbol{x}_t^i - \bar{\boldsymbol{x}}_r\| \leq 2\eta I\left(2\sigma + \frac{\gamma}{\eta}\right), \tag{8}$$

and

$$\mathbb{1}(\bar{\mathcal{A}}_r)\|\boldsymbol{x}_t^i - \bar{\boldsymbol{x}}_r\| \leq \gamma I. \tag{9}$$

*Proof of Lemma 1.* To show (8) holds, it suffices to show that under the event $\mathcal{A}_r$,

$$\|\boldsymbol{x}_t^i - \bar{\boldsymbol{x}}_r\| \le 2\eta(t - t_r)\left(2\sigma + \frac{\gamma}{\eta}\right)$$

holds for any $t_r + 1 \le t \le t_{r+1}$ and $i \in [N]$. We will show it by induction. In particular, to show that this fact holds for $t = t_r + 1$, notice

$$\|\boldsymbol{x}_{t_r+1}^i - \bar{\boldsymbol{x}}_r\| = \eta\|\boldsymbol{g}_{t_r+1}^i\| \le \eta\|\nabla F_i(\bar{\boldsymbol{x}}_r; \xi_{t_r}^i) - \boldsymbol{G}_r^i\| + \eta\|\boldsymbol{G}_r\| \le 2\eta\sigma + \gamma \le 2\eta\left(\sigma + \frac{\gamma}{\eta}\right),$$

where we used the fact that $\|\boldsymbol{G}_r\| \le \frac{\gamma}{\eta}$ under $\mathcal{A}_r$, and $\|\nabla F_i(\bar{\boldsymbol{x}}_r; \xi_{t_r}^i) - \nabla F_i(\bar{\boldsymbol{x}}_r)\| \le \sigma$, $\|\boldsymbol{G}_r^i - \nabla F_i(\bar{\boldsymbol{x}}_r)\| \le \sigma$ hold almost surely. Now, denote $\Lambda = 2\left(2\sigma + \frac{\gamma}{\eta}\right)$ and suppose that

$$\|\boldsymbol{x}_t^i - \bar{\boldsymbol{x}}_r\| \le \Lambda\eta(t - t_r). \tag{10}$$

Then we have

$$\begin{aligned}
\|\boldsymbol{x}_{t+1}^i - \bar{\boldsymbol{x}}_r\| &= \|\boldsymbol{x}_t^i - \bar{\boldsymbol{x}}_r - \eta\boldsymbol{g}_t^i\| \\
&\le \Lambda\eta(t - t_r) + \eta\|\nabla F_i(\boldsymbol{x}_t^i, \xi_t^i) - \boldsymbol{G}_r^i\| + \eta\|\boldsymbol{G}_r\| \\
&\le \Lambda\eta(t - t_r) + \eta\|\nabla f_i(\boldsymbol{x}_t^i) - \nabla f_i(\bar{\boldsymbol{x}}_r)\| + 2\eta\sigma + \gamma. \tag{11}
\end{aligned}$$

Using our assumption $\eta\Lambda I \le C/L_1$ together with the inductive assumption (10), we can apply Lemma 6 to obtain

$$\begin{aligned}
\|\nabla f_i(\boldsymbol{x}_t^i) - \nabla f_i(\bar{\boldsymbol{x}}_r)\| &\le (AL_0 + BL_1\|\nabla f_i(\bar{\boldsymbol{x}}_r)\|)\|\boldsymbol{x}_t^i - \bar{\boldsymbol{x}}_r\| \\
&\le \Lambda\eta(t - t_r)(AL_0 + BL_1\|\nabla f_i(\bar{\boldsymbol{x}}_r)\|) \\
&\overset{(i)}{\le} \Lambda\eta(t - t_r)(AL_0 + BL_1(\kappa + \rho\|\nabla f(\bar{\boldsymbol{x}}_r)\|)) \\
&\le \Lambda\eta(t - t_r)(AL_0 + BL_1\kappa) + \eta\Lambda BL_1\rho(t - t_r)(\|\nabla f(\bar{\boldsymbol{x}}_r) - \boldsymbol{G}_r\| + \|\boldsymbol{G}_r\|) \\
&\le \Lambda\eta(t - t_r)\left(AL_0 + BL_1\kappa + BL_1\rho\left(\sigma + \frac{\gamma}{\eta}\right)\right) \\
&\overset{(ii)}{\le} \frac{\Lambda(t - t_r)}{2I} \le \frac{\Lambda}{2}, \tag{12}
\end{aligned}$$

where $(i)$ comes from the heterogeneity assumption $\|\nabla f_i(x)\| \le \kappa + \rho\|\nabla f(x)\|$ for all $x$ and $(ii)$ from the assumption $2\eta I(AL_0 + BL_1\kappa + BL_1\rho(\sigma + \frac{\gamma}{\eta})) \le 1$. Substituting this into Equation (11) yields

$$\begin{aligned}
\|\boldsymbol{x}_{t+1}^i - \bar{\boldsymbol{x}}_r\| &\le \Lambda\eta(t - t_r) + \eta\frac{\Lambda}{2} + 2\eta\sigma + \gamma \\
&\le \eta\left(\Lambda(t - t_r) + \frac{\Lambda}{2} + 2\sigma + \frac{\gamma}{\eta}\right) \\
&\le \Lambda\eta(t - t_r + 1).
\end{aligned}$$

which completes the induction and the proof of Equation (8).

Next, to show Equation (9), notice that under the event $\bar{\mathcal{A}}_r$ we have

$$\|\bar{\boldsymbol{x}}_r - \boldsymbol{x}_t^i\| = \left\|\gamma\sum_{s=t_r+1}^{t-1}\frac{\boldsymbol{g}_s^i}{\|\boldsymbol{g}_s^i\|}\right\| \le \gamma\sum_{s=t_r+1}^{t-1}\left\|\frac{\boldsymbol{g}_s^i}{\|\boldsymbol{g}_s^i\|}\right\| = \gamma(t - (t_r + 1)) \le \gamma I.$$

$\square$

## B.2 PROOF OF LEMMA 2

**Lemma 2 restated.** Suppose $2\eta I(AL_0 + BL_1\kappa + BL_1\rho(\sigma + \frac{\gamma}{\eta})) \le 1$ and $\max\left\{2\eta I\left(2\sigma + \frac{\gamma}{\eta}\right), \gamma I\right\} \le \frac{C}{L_1}$. Then for all $\boldsymbol{x} \in \mathbb{R}^d$ such that $\|\boldsymbol{x} - \bar{\boldsymbol{x}}_r\| \le 2\eta I\left(2\sigma + \frac{\gamma}{\eta}\right)$, we have the following inequality almost surely holds:

$$\mathbb{1}(\mathcal{A}_r)\|\nabla^2 f_i(\boldsymbol{x})\| \le L_0 + L_1\left(\kappa + (\rho + 1)\left(\frac{\gamma}{\eta} + 2\sigma\right)\right).$$

*Proof of Lemma 2.* Under the event $\mathcal{A}_r = \{\|\boldsymbol{G}_r\| \le \gamma/\eta\}$. From the definition of $(L_0, L_1)$-smoothness we have

$$
\begin{aligned}
\|\nabla^2 f_i(\boldsymbol{x})\| &\le L_0 + L_1 \|\nabla f_i(\boldsymbol{x})\| \\
&\le L_0 + L_1 \left( \|\nabla f_i(\boldsymbol{x}) - \nabla f_i(\bar{\boldsymbol{x}}_r)\| + \|\nabla f_i(\bar{\boldsymbol{x}}_r)\| \right) \\
&\overset{(i)}{\le} L_0 + L_1 \left( \|\nabla f_i(\boldsymbol{x}) - \nabla f_i(\bar{\boldsymbol{x}}_r)\| + \kappa + \rho \|\nabla f(\bar{\boldsymbol{x}}_r)\| \right) \\
&\overset{(ii)}{\le} L_0 + L_1 \left( \|\nabla f_i(\boldsymbol{x}) - \nabla f_i(\bar{\boldsymbol{x}}_r)\| + \kappa + \rho \left( \sigma + \frac{\gamma}{\eta} \right) \right),
\end{aligned}
\tag{13}
$$

where we used the heterogeneity assumption $\|\nabla f_i(\boldsymbol{x})\| \le \kappa + \rho \|\nabla f(\bar{\boldsymbol{x}}_r)\|$ for all $\boldsymbol{x}$ to obtain $(i)$ and the fact $\|\nabla f(\bar{\boldsymbol{x}}_r)\| \le \|\nabla f(\bar{\boldsymbol{x}}_r) - \boldsymbol{G}_r\| + \|\boldsymbol{G}_r\|$ to obtain $(ii)$. Now, for all $\boldsymbol{x}$ such that $\|\boldsymbol{x} - \bar{\boldsymbol{x}}_r\| \le 2\eta I(2\sigma + \frac{\gamma}{\eta})$, according to our assumptions, we have $\|\boldsymbol{x} - \bar{\boldsymbol{x}}_r\| \le 2\eta I(2\sigma + \frac{\gamma}{\eta}) \le \frac{C}{L_1}$. Hence we can apply Lemma 6 to $\boldsymbol{x}$ and $\bar{\boldsymbol{x}}_r$, which yields

$$
\begin{aligned}
\|\nabla f_i(\boldsymbol{x}) - \nabla f_i(\bar{\boldsymbol{x}}_r)\| &\le (AL_0 + BL_1 \|\nabla f_i(\bar{\boldsymbol{x}}_r)\|) \|\boldsymbol{x} - \bar{\boldsymbol{x}}_r\| \\
&\le 2\eta I \left( 2\sigma + \frac{\gamma}{\eta} \right) (AL_0 + BL_1 \|\nabla f_i(\bar{\boldsymbol{x}}_r)\|) \\
&\le 2\eta I \left( 2\sigma + \frac{\gamma}{\eta} \right) (AL_0 + BL_1(\kappa + \rho \|\nabla f(\bar{\boldsymbol{x}}_r)\|)) \\
&\le 2\eta I \left( 2\sigma + \frac{\gamma}{\eta} \right) \left( AL_0 + BL_1 \kappa + BL_1 \rho \left( \frac{\gamma}{\eta} + \sigma \right) \right) \\
&\overset{(i)}{\le} 2\sigma + \frac{\gamma}{\eta},
\end{aligned}
$$

where $(i)$ comes from the assumption $2\eta I(AL_0 + BL_1 \kappa + BL_1 \rho(\sigma + \frac{\gamma}{\eta})) \le 1$. Substituting this result into Equation (13) yields

$$
\begin{aligned}
\|\nabla^2 f_i(\boldsymbol{x})\| &\le L_0 + L_1 \left( 2\sigma + \frac{\gamma}{\eta} + \kappa + \rho \left( \sigma + \frac{\gamma}{\eta} \right) \right) \\
&\le L_0 + L_1 \left( \kappa + (\rho + 1) \left( 2\sigma + \frac{\gamma}{\eta} \right) \right).
\end{aligned}
$$

$\square$

## B.3 PROOF OF LEMMA 3

**Lemma 3 restated.** Suppose $2\eta I(AL_0 + BL_1 \kappa + BL_1 \rho(\sigma + \frac{\gamma}{\eta})) \le 1$ and $\max \left\{ 2\eta I(2\sigma + \frac{\gamma}{\eta}), \gamma I \right\} \le \frac{C}{L_1}$, we have both

$$
\mathbb{E}_r \left[ \mathbb{1}(\mathcal{A}_r) \|\boldsymbol{x}_t^i - \bar{\boldsymbol{x}}_r\|^2 \right] \le 36 p_r I^2 \eta^2 \|\nabla f(\bar{\boldsymbol{x}}_r)\|^2 + 126 p_r I^2 \eta^2 \sigma^2,
\tag{14}
$$

$$
\mathbb{E}_r \left[ \mathbb{1}(\mathcal{A}_r) \|\boldsymbol{x}_t^i - \bar{\boldsymbol{x}}_r\|^2 \right] \le 18 p_r I^2 \eta \gamma \|\nabla f(\bar{\boldsymbol{x}}_r)\| + 18 p_r I^2 \eta^2 \left( \frac{\gamma}{\eta} \sigma + 5\sigma^2 \right),
\tag{15}
$$

hold for any $t - 1 \in \mathcal{I}_r$.

*Proof of Lemma 3.* Under the event $\mathcal{A}_r$, the local update rule is given by

$$
\boldsymbol{x}_{t+1}^i = \boldsymbol{x}_t^i - \eta \boldsymbol{g}_t^i, \quad \text{where} \quad \boldsymbol{g}_t^i = \nabla F_i(\boldsymbol{x}_t^i; \xi_t^i) - \boldsymbol{G}_r^i + \boldsymbol{G}_r.
$$

Using the basic inequality $(a + b)^2 \le (1 + 1/\lambda)a^2 + (\lambda + 1)b^2$ for any $\lambda > 0$, we have

$$
\begin{aligned}
&\mathbb{E}_r \left[ \mathbb{1}(\mathcal{A}_r) \|\boldsymbol{x}_{t+1}^i - \bar{\boldsymbol{x}}_r\|^2 \right] \\
&= \mathbb{E}_r \left[ \mathbb{1}(\mathcal{A}_r) \|\boldsymbol{x}_t^i - \bar{\boldsymbol{x}}_r - \eta \boldsymbol{g}_t^i\|^2 \right] \\
&\overset{(i)}{\le} \mathbb{E}_r \left[ \mathbb{1}(\mathcal{A}_r) \|\boldsymbol{x}_t^i - \bar{\boldsymbol{x}}_r - \eta(\nabla f_i(\boldsymbol{x}_t^i) - \boldsymbol{G}_r^i + \boldsymbol{G}_r)\|^2 \right] + \eta^2 \mathbb{E}_r \left[ \mathbb{1}(\mathcal{A}_r) \|\nabla F_i(\boldsymbol{x}_t^i; \xi_t^i) - \nabla f_i(\boldsymbol{x}_t^i)\|^2 \right] \\
&\overset{(ii)}{\le} \left( \frac{1}{I} + 1 \right) \mathbb{E}_r \left[ \mathbb{1}(\mathcal{A}_r) \|\boldsymbol{x}_t^i - \bar{\boldsymbol{x}}_r\|^2 \right] + (I + 1)\eta^2 \mathbb{E}_r \left[ \mathbb{1}(\mathcal{A}_r) \|\nabla f_i(\boldsymbol{x}_t^i) - \boldsymbol{G}_r^i + \boldsymbol{G}_r\|^2 \right] + p_r \eta^2 \sigma^2.
\end{aligned}
\tag{16}
$$

The equality $(i)$ and $(ii)$ hold since $\mathcal{F}_r \subseteq \mathcal{H}_t$ for $t \geq t_r$ such that

$$
\begin{aligned}
&\mathbb{E}_r \left[ \mathbb{1}(\mathcal{A}_r) \left\langle \boldsymbol{x}_t^i - \bar{\boldsymbol{x}}_r - \eta(\nabla f_i(\boldsymbol{x}_t^i) - \boldsymbol{G}_r^i + \boldsymbol{G}_r), \nabla F_i(\boldsymbol{x}_t^i; \xi_t^i) - \nabla f_i(\boldsymbol{x}_t^i) \right\rangle \right] \\
=&\mathbb{E}_r \left[ \mathbb{E} \left[ \mathbb{1}(\mathcal{A}_r) \left\langle \boldsymbol{x}_t^i - \bar{\boldsymbol{x}}_r - \eta(\nabla f_i(\boldsymbol{x}_t^i) - \boldsymbol{G}_r^i + \boldsymbol{G}_r), \nabla F_i(\boldsymbol{x}_t^i; \xi_t^i) - \nabla f_i(\boldsymbol{x}_t^i) \right\rangle \big| \mathcal{H}_t \right] \right] \\
=&\mathbb{E}_r \left[ \mathbb{1}(\mathcal{A}_r) \left\langle \boldsymbol{x}_t^i - \bar{\boldsymbol{x}}_r - \eta(\nabla f_i(\boldsymbol{x}_t^i) - \boldsymbol{G}_r^i + \boldsymbol{G}_r), \mathbb{E} \left[ \nabla F_i(\boldsymbol{x}_t^i; \xi_t^i) - \nabla f_i(\boldsymbol{x}_t^i) \big| \mathcal{H}_t \right] \right\rangle \right] = 0,
\end{aligned}
$$

and

$$
\begin{aligned}
\mathbb{E}_r \left[ \mathbb{1}(\mathcal{A}_r) \| \nabla F_i(\boldsymbol{x}_t^i; \xi_t^i) - \nabla f_i(\boldsymbol{x}_t^i) \|^2 \right] &= \mathbb{E}_r \left[ \mathbb{E} \left[ \| \nabla F_i(\boldsymbol{x}_t^i; \xi_t^i) - \nabla f_i(\boldsymbol{x}_t^i) \|^2 \big| \mathcal{H}_t \right] \right] \\
&\leq \mathbb{E}_r \left[ \mathbb{1}(\mathcal{A}_r) \sigma^2 \right] = p_r \sigma^2.
\end{aligned}
$$

Let $L = L_0 + L_1(\kappa + (\rho+1)(\frac{\gamma}{\eta} + 2\sigma))$. Applying the upper bound for Hessian matrix in Lemma 2 and the premise in Lemma 1, we have

$$
\begin{aligned}
&\mathbb{E}_r \left[ \mathbb{1}(\mathcal{A}_r) \| \nabla f_i(\boldsymbol{x}_t^i) - \boldsymbol{G}_r^i + \boldsymbol{G}_r \|^2 \right] \\
&= \mathbb{E}_r \left[ \mathbb{1}(\mathcal{A}_r) \left\| (\nabla f_i(\boldsymbol{x}_t^i) - \nabla f_i(\bar{\boldsymbol{x}}_r)) + (\nabla f_i(\bar{\boldsymbol{x}}_r) - \boldsymbol{G}_r^i) + \boldsymbol{G}_r \right\|^2 \right] \\
&\leq 2\mathbb{E}_r \left[ \mathbb{1}(\mathcal{A}_r) \left\| (\nabla f_i(\boldsymbol{x}_t^i) - \nabla f_i(\bar{\boldsymbol{x}}_r)) + (\nabla f_i(\bar{\boldsymbol{x}}_r) - \boldsymbol{G}_r^i) \right\|^2 \right] + 2\mathbb{E}_r \left[ \mathbb{1}(\mathcal{A}_r) \| \boldsymbol{G}_r \|^2 \right] \\
&\leq 4\mathbb{E}_r \left[ \mathbb{1}(\mathcal{A}_r) \| \nabla f_i(\boldsymbol{x}_t^i) - \nabla f_i(\bar{\boldsymbol{x}}_r) \|^2 \right] + 4 p_r \sigma^2 + 2\mathbb{E}_r \left[ \mathbb{1}(\mathcal{A}_r) \| \boldsymbol{G}_r \|^2 \right] \\
&\leq 4\mathbb{E}_r \left[ \mathbb{1}(\mathcal{A}_r) \left\| \int_0^1 \nabla^2 f_i(\alpha \boldsymbol{x}_t^i + (1-\alpha)\bar{\boldsymbol{x}}_r)(\boldsymbol{x}_t^i - \bar{\boldsymbol{x}}_r) d\alpha \right\|^2 \right] + 4 p_r \sigma^2 + 2\mathbb{E}_r \left[ \mathbb{1}(\mathcal{A}_r) \| \boldsymbol{G}_r \|^2 \right] \\
&\leq 4 L^2 \mathbb{E}_r \left[ \mathbb{1}(\mathcal{A}_r) \| \boldsymbol{x}_t^i - \bar{\boldsymbol{x}}_r \|^2 \right] + 4 p_r \sigma^2 + 2\mathbb{E}_r \left[ \mathbb{1}(\mathcal{A}_r) \| \boldsymbol{G}_r \|^2 \right], \quad (17)
\end{aligned}
$$

where the second inequality follows from $\| \boldsymbol{G}_r^i - \nabla f_i(\bar{\boldsymbol{x}}_r) \| \leq \sigma$ almost surely. Plugging the final bound of (17) into (16) yields

$$
\begin{aligned}
\mathbb{E}_r \left[ \mathbb{1}(\mathcal{A}_r) \| \boldsymbol{x}_{t+1}^i - \bar{\boldsymbol{x}}_r \|^2 \right] \leq &\left( \frac{1}{I} + 1 + 4 L I \eta^2 \right) \mathbb{E}_r \left[ \mathbb{1}(\mathcal{A}_r) \| \boldsymbol{x}_t^i - \bar{\boldsymbol{x}}_r \|^2 \right] \\
&+ 2(I+1)\eta^2 \mathbb{E}_r \left[ \mathbb{1}(\mathcal{A}_r) \| \boldsymbol{G}_r \|^2 \right] + 10 p_r (I+1)\eta^2 \sigma^2. \quad (18)
\end{aligned}
$$

By recursively invoking (18), we are guaranteed that

$$
\begin{aligned}
\mathbb{E}_r \left[ \mathbb{1}(\mathcal{A}_r) \| \boldsymbol{x}_{t+1}^i - \bar{\boldsymbol{x}}_r \|^2 \right] &\leq \sum_{s=0}^{I-1} \left( \frac{1}{I} + 1 + 4 L I \eta^2 \right)^s (I+1) \left( 2\eta^2 \mathbb{E}_r \left[ \mathbb{1}(\mathcal{A}_r) \| \boldsymbol{G}_r \|^2 \right] + 10 p_r \eta^2 \sigma^2 \right) \\
&= \frac{\left( \frac{1}{I} + 1 + 4 L I \eta^2 \right)^I}{\frac{1}{I} + 4 L I \eta^2} (I+1) \left( 2\eta^2 \mathbb{E}_r \left[ \mathbb{1}(\mathcal{A}_r) \| \boldsymbol{G}_r \|^2 \right] + 10 p_r \eta^2 \sigma^2 \right) \\
&\overset{(i)}{\leq} \frac{\left( \frac{2}{I} + 1 \right)^I}{\frac{1}{I}} (I+1) \left( 2\eta^2 \mathbb{E}_r \left[ \mathbb{1}(\mathcal{A}_r) \| \boldsymbol{G}_r \|^2 \right] + 10 p_r \eta^2 \sigma^2 \right) \\
&\overset{(ii)}{\leq} 9 \left( 2 I^2 \eta^2 \mathbb{E}_r \left[ \mathbb{1}(\mathcal{A}_r) \| \boldsymbol{G}_r \|^2 \right] + 10 p_r I^2 \eta^2 \sigma^2 \right) \\
&\leq 36 I^2 \eta^2 \left( \mathbb{E}_r \left[ \mathbb{1}(\mathcal{A}_r) \| \boldsymbol{G}_r - \nabla f(\bar{\boldsymbol{x}}_r) \|^2 \right] + p_r \| \nabla f(\bar{\boldsymbol{x}}_r) \|^2 \right) + 90 p_r I^2 \eta^2 \sigma^2 \\
&\overset{(iii)}{\leq} 36 p_r I^2 \eta^2 \| \nabla f(\bar{\boldsymbol{x}}_r) \|^2 + 126 p_r I^2 \eta^2 \sigma^2.
\end{aligned}
$$

The inequality $(i)$ comes from

$$
4 I \eta^2 L^2 = \frac{1}{I} (2 I \eta L)^2 \leq \frac{1}{I} \left( 2 I \eta \left( L_0 + L_1 \kappa + L_1 (\rho+1)(2\sigma + \frac{\gamma}{\eta}) \right) \right)^2 \leq \frac{1}{I},
$$

which is true because $2\eta I(A L_0 + B L_1 \kappa + B L_1 \rho(\sigma + \frac{\gamma}{\eta})) \leq 1$ and $A, B \geq 1$. The inequality $(ii)$ comes from $(\frac{2}{I} + 1)^I (I+1) \leq e^2 I$ for any $I \geq 1$. The inequality $(iii)$ holds since $\| \boldsymbol{G}_r - \nabla f(\bar{\boldsymbol{x}}_r) \| \leq$

$\sigma$ almost surely. Therefore, we have proved (14). In addition, for (15), we notice that

$$
\begin{aligned}
\mathbb{E}_r \left[ \mathbb{1}(\mathcal{A}_r) \| \boldsymbol{x}_{t+1}^i - \bar{\boldsymbol{x}}_r \|^2 \right] &\leq 18 I^2 \eta^2 \mathbb{E}_r \left[ \mathbb{1}(\mathcal{A}_r) \| \boldsymbol{G}_r \|^2 \right] + 90 p_r I^2 \eta^2 \sigma^2 \\
&\leq 18 I^2 \eta^2 \mathbb{E}_r \left[ \mathbb{1}(\mathcal{A}_r) \| \boldsymbol{G}_r \| \left( \| \boldsymbol{G}_r - \nabla f(\bar{\boldsymbol{x}}_r) \| + \| \nabla f(\bar{\boldsymbol{x}}_r) \| \right) \right] + 90 p_r I^2 \eta^2 \sigma^2 \\
&\overset{(iv)}{\leq} 18 p_r I^2 \eta^2 \frac{\gamma}{\eta} \left( \sigma + \| \nabla f(\bar{\boldsymbol{x}}_r) \| \right) + 90 p_r I^2 \eta^2 \sigma^2 \\
&= 18 p_r I^2 \eta \gamma \| \nabla f(\bar{\boldsymbol{x}}_r) \| + 18 p_r I^2 \eta^2 \left( \frac{\gamma}{\eta} \sigma + 5 \sigma^2 \right).
\end{aligned}
$$

The inequality $(iv)$ holds since $\| \boldsymbol{G}_r \| \leq \gamma / \eta$ holds under the event $\mathcal{A}_r$ and $\| \boldsymbol{G}_r - \nabla f(\bar{\boldsymbol{x}}_r) \| \leq \sigma$ almost surely.

$\square$

## C  PROOF OF MAIN RESULTS

### C.1  PROOF OF LEMMA 4

**Lemma 4 restated.**  Under the conditions of Lemma 1, let $p_r = \mathbb{P}(\mathcal{A}_r | \mathcal{F}_r)$, $\Gamma = A L_0 + B L_1 (\kappa + \rho(\frac{\gamma}{\eta} + \sigma))$. Then it holds that for each $1 \leq r \leq R - 1$,

$$
\mathbb{E}_r \left[ f(\bar{\boldsymbol{x}}_{r+1}) - f(\bar{\boldsymbol{x}}_r) \right] \leq \mathbb{E}_r \left[ \mathbb{1}(\mathcal{A}_r) V(\bar{\boldsymbol{x}}_r) \right] + \mathbb{E}_r \left[ \mathbb{1}(\bar{\mathcal{A}}_r) U(\bar{\boldsymbol{x}}_r) \right],
$$

where

$$
\begin{aligned}
V(\bar{\boldsymbol{x}}_r) = & \left( -\frac{\eta I}{2} + 36 \Gamma^2 I^3 \eta^3 + 9 \frac{\gamma}{\eta} B L_1 I^2 \eta^2 \right) \| \nabla f(\bar{\boldsymbol{x}}_r) \|^2 + 9 B L_1 I^2 \eta^2 \left( 5 \sigma^2 + \frac{\gamma}{\eta} \sigma \right) \| \nabla f(\bar{\boldsymbol{x}}_r) \| \\
& + 90 \Gamma^2 I^3 \eta^3 \sigma^2 + \frac{2 A L_0 I \eta^2 \sigma^2}{N},
\end{aligned}
$$

and

$$
U(\bar{\boldsymbol{x}}_r) = \left( -\frac{2}{5} \gamma I + \frac{B L_1 (4 \rho + 1) \gamma^2 I^2}{2} \right) \| \nabla f(\bar{\boldsymbol{x}}_r) \| - \frac{3 \gamma^2 I}{5 \eta} + \gamma^2 I^2 (3 A L_0 + 2 B L_1 \kappa) + 6 \gamma I \sigma.
$$

*Proof.*  We begin by applying Lemma 5 to obtain a bound on $f(\bar{\boldsymbol{x}}_{r+1}) - f(\bar{\boldsymbol{x}}_r)$, but first we must show that the conditions of Lemma 5 hold here. Note that

$$
\begin{aligned}
\| \bar{\boldsymbol{x}}_{r+1} - \bar{\boldsymbol{x}}_r \| &= \left\| \frac{1}{N} \sum_{i=1}^N \boldsymbol{x}_{t_{r+1}}^i - \bar{\boldsymbol{x}}_r \right\| \\
&\leq \frac{1}{N} \sum_{i=1}^N \mathbb{1}(\mathcal{A}_r) \| \boldsymbol{x}_{t_{r+1}}^i - \bar{\boldsymbol{x}}_r \| + \frac{1}{N} \sum_{i=1}^N \mathbb{1}(\bar{\mathcal{A}}_r) \| \boldsymbol{x}_{t_{r+1}}^i - \bar{\boldsymbol{x}}_r \| \\
&\leq \max \left\{ 2 \eta I \left( 2 \sigma + \frac{\gamma}{\eta} \right), \gamma I \right\} \leq \frac{C}{L_1},
\end{aligned}
$$

where the last step is due to the conditions of Lemma 1. This shows that we can apply Lemma 5 to obtain

$$
\begin{aligned}
\mathbb{E}_r \left[ f(\bar{\boldsymbol{x}}_{r+1}) - f(\bar{\boldsymbol{x}}_r) \right] \leq {} & \mathbb{E}_r \left[ \langle \nabla f(\bar{\boldsymbol{x}}_r), \bar{\boldsymbol{x}}_{r+1} - \bar{\boldsymbol{x}}_r \rangle \right] + \mathbb{E}_r \left[ \frac{A L_0 + B L_1 \| \nabla f(\bar{\boldsymbol{x}}_r) \|}{2} \| \bar{\boldsymbol{x}}_{r+1} - \bar{\boldsymbol{x}}_r \|^2 \right] \\
\leq {} & -\eta \mathbb{E}_r \left[ \frac{1}{N} \sum_{i=1}^N \sum_{t \in \mathcal{I}_r} \mathbb{1}(\mathcal{A}_r) \langle \nabla f(\bar{\boldsymbol{x}}_r), \boldsymbol{g}_t^i \rangle \right] \\
& - \gamma \mathbb{E}_r \left[ \frac{1}{N} \sum_{i=1}^N \sum_{t \in \mathcal{I}_r} \mathbb{1}(\bar{\mathcal{A}}_r) \langle \nabla f(\bar{\boldsymbol{x}}_r), \frac{\boldsymbol{g}_t^i}{\| \boldsymbol{g}_t^i \|} \rangle \right] \\
& + \frac{A L_0}{2} \mathbb{E}_r \left[ \| \bar{\boldsymbol{x}}_{r+1} - \bar{\boldsymbol{x}}_r \|^2 \right] + \frac{B L_1}{2} \| \nabla f(\bar{\boldsymbol{x}}_r) \| \mathbb{E}_r \left[ \| \bar{\boldsymbol{x}}_{r+1} - \bar{\boldsymbol{x}}_r \|^2 \right].
\end{aligned}
$$

$$(19)$$

Let $p_r = \mathbb{P}(\mathcal{A}_r | \mathcal{F}_r)$, then $1 - p_r = \mathbb{P}(\bar{\mathcal{A}}_r | \mathcal{F}_r)$. Notice that $p_r$ is a function of $\bar{\boldsymbol{x}}_r$. The last term in Equation (19) can be bounded as follows:

$$
\|\nabla f(\bar{\boldsymbol{x}}_r)\| \mathbb{E}_r \left[ \|\bar{\boldsymbol{x}}_{r+1} - \bar{\boldsymbol{x}}_r\|^2 \right]
$$
$$
= \|\nabla f(\bar{\boldsymbol{x}}_r)\| \mathbb{E} \left[ \mathbb{1}(\mathcal{A}_r) \|\bar{\boldsymbol{x}}_{r+1} - \bar{\boldsymbol{x}}_r\|^2 \big| \mathcal{F}_r \right] + \|\nabla f(\bar{\boldsymbol{x}}_r)\| \mathbb{E} \left[ \mathbb{1}(\bar{\mathcal{A}}_r) \|\bar{\boldsymbol{x}}_{r+1} - \bar{\boldsymbol{x}}_r\|^2 \big| \mathcal{F}_r \right]
$$
$$
\overset{(i)}{\leq} \|\nabla f(\bar{\boldsymbol{x}}_r)\| \mathbb{E} \left[ \mathbb{1}(\mathcal{A}_r) \|\bar{\boldsymbol{x}}_{r+1} - \bar{\boldsymbol{x}}_r\|^2 \big| \mathcal{F}_r \right] + (1 - p_r) \gamma^2 I^2 \|\nabla f(\bar{\boldsymbol{x}}_r)\|
$$
$$
\overset{(ii)}{\leq} 18 p_r I^2 \eta^2 \|\nabla f(\bar{\boldsymbol{x}}_r)\| \left( \frac{\gamma}{\eta} \|\nabla f(\bar{\boldsymbol{x}}_r)\| + 5\sigma^2 + \frac{\gamma}{\eta}\sigma \right) + (1 - p_r) \gamma^2 I^2 \|\nabla f(\bar{\boldsymbol{x}}_r)\|
$$
$$
\leq 18 p_r I^2 \eta \gamma \|\nabla f(\bar{\boldsymbol{x}}_r)\|^2 + 18 p_r I^2 \eta^2 \left( 5\sigma^2 + \frac{\gamma}{\eta}\sigma \right) \|\nabla f(\bar{\boldsymbol{x}}_r)\| + (1 - p_r) \gamma^2 I^2 \|\nabla f(\bar{\boldsymbol{x}}_r)\|, \quad (20)
$$

where $(i)$ comes from an application of Lemma 1 with $t = t_{r+1}$, and $(ii)$ comes from an application of (15) in Lemma 3. Substituting (20) into (19) gives

$$
\mathbb{E}_r \left[ f(\bar{\boldsymbol{x}}_{r+1}) - f(\bar{\boldsymbol{x}}_r) \right]
$$
$$
\leq -\eta \mathbb{E}_r \left[ \frac{1}{N} \sum_{i=1}^N \sum_{t \in \mathcal{I}_r} \mathbb{1}(\mathcal{A}_r) \langle \nabla f(\bar{\boldsymbol{x}}_r), \boldsymbol{g}_t^i \rangle \right] - \gamma \mathbb{E}_r \left[ \frac{1}{N} \sum_{i=1}^N \sum_{t \in \mathcal{I}_r} \mathbb{1}(\bar{\mathcal{A}}_r) \langle \nabla f(\bar{\boldsymbol{x}}_r), \frac{\boldsymbol{g}_t^i}{\|\boldsymbol{g}_t^i\|} \rangle \right]
$$
$$
+ \frac{AL_0}{2} \mathbb{E}_r \left[ \|\bar{\boldsymbol{x}}_{r+1} - \bar{\boldsymbol{x}}_r\|^2 \right] + 9 p_r BL_1 I^2 \eta^2 \left( \frac{\gamma}{\eta} \|\nabla f(\bar{\boldsymbol{x}}_r)\|^2 + \left( 5\sigma^2 + \frac{\gamma}{\eta}\sigma \right) \|\nabla f(\bar{\boldsymbol{x}}_r)\| \right)
$$
$$
+ (1 - p_r) \frac{BL_1 \gamma^2 I^2}{2} \|\nabla f(\bar{\boldsymbol{x}}_r)\| \quad (21)
$$

We introduce three claims to bound the first three terms in (21), whose proofs are deferred to Section D.

**Claim 1.** *Under the conditions of Lemma 4, we have*

$$
-\gamma \mathbb{E}_r \left[ \frac{1}{N} \sum_{i=1}^N \sum_{t \in \mathcal{I}_r} \mathbb{1}(\bar{\mathcal{A}}_r) \langle \nabla f(\bar{\boldsymbol{x}}_r), \frac{\boldsymbol{g}_t^i}{\|\boldsymbol{g}_t^i\|} \rangle \right]
$$
$$
\leq (1 - p_r) \left[ \left( -\frac{2}{5}\gamma I + 2BL_1 \rho \gamma^2 I^2 \right) \|\nabla f(\bar{\boldsymbol{x}}_r)\| - \frac{3\gamma^2 I}{5\eta} + 2\gamma^2 I^2 (AL_0 + BL_1 \kappa) + 6\gamma I \sigma \right].
$$

**Claim 2.** *Under the conditions of Lemma 4, we have*

$$
-\eta \mathbb{E}_r \left[ \frac{1}{N} \sum_{i=1}^N \sum_{t \in \mathcal{I}_r} \mathbb{1}(\mathcal{A}_r) \langle \nabla f(\bar{\boldsymbol{x}}_r), \boldsymbol{g}_t^i \rangle \right]
$$
$$
\leq p_r \left[ \left( -\frac{\eta I}{2} + 36 I^3 \eta^3 \Gamma^2 \right) \|\nabla f(\bar{\boldsymbol{x}}_r)\|^2 + 126 I^3 \eta^3 \sigma^2 \Gamma^2 \right] - \frac{\eta}{2I} \mathbb{E}_r \left[ \mathbb{1}(\mathcal{A}_r) \left\| \frac{1}{N} \sum_{i=1}^N \sum_{t \in \mathcal{I}_r} \nabla f(\boldsymbol{x}_t^i) \right\|^2 \right],
$$

*where $\Gamma = AL_0 + BL_1 \left( \kappa + \rho \left( \sigma + \frac{\gamma}{\eta} \right) \right)$.*

**Claim 3.** *Under the conditions of Lemma 4, we have*

$$
\mathbb{E}_r \left[ \|\bar{\boldsymbol{x}}_{r+1} - \bar{\boldsymbol{x}}_r\|^2 \right] \leq 2(1 - p_r) \gamma^2 I^2 + \frac{4 p_r I \sigma^2 \eta^2}{N} + 4\eta^2 \mathbb{E}_r \left[ \mathbb{1}(\mathcal{A}_r) \left\| \frac{1}{N} \sum_{i=1}^N \sum_{t \in \mathcal{I}_r} \nabla f_i(\boldsymbol{x}_t^i) \right\|^2 \right].
$$

Combining Claims 1, 2, and 3 with (19) and (20) yields

$$\mathbb{E}_r\left[f(\bar{\boldsymbol{x}}_{r+1}) - f(\bar{\boldsymbol{x}}_r)\right]$$

$$\leq p_r\left[\left(-\frac{\eta I}{2} + 36\Gamma^2 I^3\eta^3 + 9\frac{\gamma}{\eta}BL_1 I^2\eta^2\right)\|\nabla f(\bar{\boldsymbol{x}}_r)\|^2 + 9p_r BL_1 I^2\eta^2\left(5\sigma^2 + \frac{\gamma}{\eta}\sigma\right)\|\nabla f(\bar{\boldsymbol{x}}_r)\| +\right.$$

$$\left. 126\Gamma^2 I^3\eta^3\sigma^2 + \frac{2AL_0 I\eta^2\sigma^2}{N}\right]$$

$$+ (1 - p_r)\left[\left(-\frac{2}{5}\gamma I + \frac{BL_1(4\rho + 1)\gamma^2 I^2}{2}\right)\|\nabla f(\bar{\boldsymbol{x}}_r)\| - \frac{3\gamma^2 I}{5\eta} + \gamma^2 I^2(3AL_0 + 2BL_1\kappa) + 6\gamma I\sigma\right]$$

$$+ \left(2AL_0\eta^2 - \frac{\eta}{2I}\right)\mathbb{E}_r\left[\mathbb{1}(\mathcal{A}_r)\left\|\frac{1}{N}\sum_{i=1}^N\sum_{t\in\mathcal{I}_r}\nabla f(\boldsymbol{x}_t^i)\right\|^2\right]$$

$$\leq p_r\left[\left(-\frac{\eta I}{2} + 36\Gamma^2 I^3\eta^3 + 9\frac{\gamma}{\eta}BL_1 I^2\eta^2\right)\|\nabla f(\bar{\boldsymbol{x}}_r)\|^2 + 9BL_1 I^2\eta^2\left(5\sigma^2 + \frac{\gamma}{\eta}\sigma\right)\|\nabla f(\bar{\boldsymbol{x}}_r)\| +\right.$$

$$\left. 90\Gamma^2 I^3\eta^3\sigma^2 + \frac{2AL_0 I\eta^2\sigma^2}{N}\right]$$

$$+ (1 - p_r)\left[\left(-\frac{2}{5}\gamma I + \frac{BL_1(4\rho + 1)\gamma^2 I^2}{2}\right)\|\nabla f(\bar{\boldsymbol{x}}_r)\| - \frac{3\gamma^2 I}{5\eta} + \gamma^2 I^2(3AL_0 + 2BL_1\kappa) + 6\gamma I\sigma\right],$$

where the last inequality holds since $\eta/(2I) \geq 4\eta^2$ due to the assumption $4AL_0\eta I \leq 1$. Then we can finish the proof of Lemma 4 by noticing that $p_r = \mathbb{E}_r[\mathbb{1}(\mathcal{A}_r)]$ and $1 - p_r = \mathbb{E}_r[\mathbb{1}(\bar{\mathcal{A}}_r)]$. □

## C.2 PROOF OF THEOREM 1

**Theorem 1 restated.** Suppose Assumption 1 hold. For any $\epsilon \leq \frac{3AL_0}{5BL_1\rho}$, we choose

$$\eta \leq \min\left\{\frac{1}{856\Gamma I}, \frac{\epsilon}{180\Gamma I\sigma}, \frac{N\epsilon^2}{8AL_0\sigma^2}\right\} \quad \text{and} \quad \gamma = \left(11\sigma + \frac{AL_0}{BL_1\rho}\right)\eta, \tag{22}$$

where $\Gamma = AL_0 + BL_1\kappa + BL_1\rho\left(\sigma + \frac{\gamma}{\eta}\right)$. The output of EPISODE satisfies

$$\frac{1}{R}\sum_{t=0}^R \mathbb{E}\left[\|\nabla f(\bar{\boldsymbol{x}}_r)\|\right] \leq 3\epsilon$$

as long as $R \geq \frac{4\Delta}{\epsilon^2\eta I}$.

*Proof.* In order to apply Lemma 4, we must verify the conditions of Lemma 1 under our choice of hyperparameters. From our choices of $\eta$ and $\gamma$, we have

$$2\Gamma\eta I \leq \frac{1}{856} < 1.$$

Also

$$2\eta I\left(2\sigma + \frac{\gamma}{\eta}\right) \overset{(i)}{\leq} \frac{2\sigma + \frac{\gamma}{\eta}}{856\left(AL_0 + BL_1\kappa + BL_1\rho\left(\sigma + \frac{\gamma}{\eta}\right)\right)} \overset{(ii)}{\leq} \frac{C}{L_1},$$

where $(i)$ comes from the condition $\eta \leq 1/(856\Gamma I)$ in (22), $(ii)$ is true due to the fact that $B, C \geq 1$ and $\rho \geq 1$. Lastly, it also holds that

$$\gamma I \leq 4\eta I\sigma + 2\gamma I = 2\eta I\left(2\sigma + \frac{\gamma}{\eta}\right) \leq \frac{C}{L_1}.$$

Therefore the conditions of Lemma 1 are satisfied, and we can apply Lemma 4. Denoting

$$U(\boldsymbol{x}) = \left(-\frac{2}{5}\gamma I + \frac{BL_1(4\rho + 1)\gamma^2 I^2}{2}\right)\|\nabla f(\boldsymbol{x})\| - \frac{3\gamma^2 I}{5\eta} + \gamma^2 I^2(3AL_0 + 2BL_1\kappa) + 6\gamma I\sigma, \tag{23}$$

and

$$V(\boldsymbol{x}) = \left(-\frac{\eta I}{2} + 36\Gamma^2 I^3 \eta^3 + 9\frac{\gamma}{\eta}I^2\eta^2\right)\|\nabla f(\boldsymbol{x})\|^2 + 9p_r I^2 \eta^2 \left(5\sigma^2 + \frac{\gamma}{\eta}\sigma\right)\|\nabla f(\boldsymbol{x})\|$$
$$+ 126\Gamma^2 I^3 \eta^3 \sigma^2 + \frac{2AL_0 I\eta^2\sigma^2}{N}. \tag{24}$$

Lemma 4 tells us that

$$\mathbb{E}_r\left[f(\bar{\boldsymbol{x}}_{r+1}) - f(\bar{\boldsymbol{x}}_r)\right] \le \mathbb{E}_r\left[\mathbb{1}(\bar{\mathcal{A}}_r)U(\bar{\boldsymbol{x}}_r) + \mathbb{1}(\mathcal{A}_r)V(\bar{\boldsymbol{x}}_r)\right]. \tag{25}$$

We will proceed by bounding each $U(\boldsymbol{x})$ and $V(\boldsymbol{x})$ by the same linear function of $\|\nabla f(\boldsymbol{x})\|$.

To bound $U(\boldsymbol{x})$, notice

$$-\frac{2}{5}\gamma I + \frac{BL_1(4\rho+1)\gamma^2 I^2}{2}$$
$$= -\frac{2}{5}\gamma I + 2BL_1\rho\gamma^2 I^2 + \frac{1}{2}BL_1\gamma^2 I^2$$
$$\le \gamma I\left(-\frac{2}{5} + 2BL_1\rho\gamma I + \frac{1}{2}BL_1\gamma I\right)$$
$$\le \gamma I\left(-\frac{2}{5} + 2BL_1\rho\left(11\sigma + \frac{AL_0}{BL_1\rho}\right)\eta I + \frac{1}{2}BL_1\left(11\sigma + \frac{AL_0}{BL_1\rho}\right)\eta I\right)$$
$$\overset{(i)}{\le} \gamma I\left(-\frac{2}{5} + 3\left(11BL_1\rho\sigma + AL_0\right)\eta I\right)$$
$$\overset{(ii)}{\le} \gamma I\left(-\frac{2}{5} + \frac{18}{856}\right) \le -\frac{3}{10}\gamma I$$
$$\overset{(iii)}{\le} -\frac{3}{10}\frac{AL_0}{BL_1\rho}\eta I \overset{(iv)}{\le} -\frac{1}{2}\epsilon\eta I, \tag{26}$$

where $(i)$ comes from $\rho \ge 1$ and $(ii)$ comes from $856\Gamma\eta I \le 1$ and $(iii)$ holds since $\gamma/\eta = 11\sigma + \frac{AL_0}{BL_1\rho}$ and $(iv)$ comes from $\epsilon \le \frac{3AL_0}{5BL_1\rho}$. Also, we have

$$-\frac{3\gamma^2 I}{5\eta} + \gamma^2 I^2(3AL_0 + 2BL_1\kappa) + 6\gamma I\sigma \le \frac{\gamma^2 I}{\eta}\left(-\frac{3}{5} + 3\Gamma\eta I + 6\sigma\frac{\eta}{\gamma}\right)$$
$$\le \frac{\gamma^2 I}{\eta}\left(-\frac{3}{5} + \frac{3}{856} + \frac{6\sigma}{11\sigma + \frac{AL_0}{BL_1\rho}}\right)$$
$$\le \frac{\gamma^2 I}{\eta}\left(-\frac{3}{5} + \frac{3}{856} + \frac{6}{11}\right) \le 0. \tag{27}$$

Plugging Equations (26) and (27) into Equation (23) yields

$$U(\boldsymbol{x}) \le -\frac{1}{2}\epsilon\eta I\|\nabla f(\boldsymbol{x})\|. \tag{28}$$

Now to bound $V(\boldsymbol{x})$, we have

$$-\frac{\eta I}{2} + 36\Gamma^2 I^3 \eta^3 + 9\frac{\gamma}{\eta}BL_1 I^2 \eta^2 \overset{(i)}{\le} -\frac{1}{2}\eta I + \frac{36}{856^2}\eta I + \frac{9(11BL_1\sigma + AL_0/\rho)}{856\Gamma}\eta I$$
$$\le -\frac{1}{4}\eta I, \tag{29}$$

where $(i)$ comes from $\eta \le \frac{1}{856\Gamma I}$ and $\Gamma > BL_1\sigma + AL_0/\rho$ for $\rho > 1$. Using the assumption $\eta \le \frac{\epsilon}{180I\Gamma\sigma}$, it holds that

$$9BL_1 I^2 \eta^2\left(5\sigma^2 + \frac{\gamma}{\eta}\sigma\right) = 9BL_1 I^2 \eta^2\left(16\sigma^2 + \frac{AL_0\sigma}{BL_1\rho}\right)$$
$$\le \eta I\epsilon\frac{16BL_1\sigma + AL_0}{20\Gamma} \overset{(ii)}{\le} \frac{1}{4}\epsilon\eta I \tag{30}$$

where $(ii)$ comes from $16BL_1\sigma + AL_0 < 5\Gamma$. Lastly, we have

$$
\begin{aligned}
90\Gamma^2 I^3 \eta^3 \sigma^2 + \frac{2AL_0 I \eta^2 \sigma^2}{N} &= \eta I \left( 90\Gamma^2 I^2 \eta^2 \sigma^2 + \frac{2AL_0 \eta \sigma^2}{N} \right) \\
&\overset{(iii)}{\leq} \eta I \left( 90\Gamma^2 \sigma^2 \cdot \frac{\epsilon^2}{180^2 \Gamma^2 \sigma^2} + \frac{2AL_0 \sigma^2}{N} \frac{N\epsilon^2}{8AL_0 \sigma^2} \right) \\
&\leq \frac{1}{4}\epsilon^2 \eta I,
\end{aligned}
\tag{31}
$$

where $(i)$ comes from $\eta \leq \min \left\{ \frac{\epsilon}{180 I \Gamma \sigma}, \frac{N\epsilon^2}{8AL_0 \sigma^2} \right\}$. Plugging Equations (29), (30), and (31) into (24) then yields

$$
V(\mathbf{x}) \leq -\frac{1}{4}\eta I \|\nabla f(\mathbf{x})\|^2 + \frac{1}{4}\epsilon \eta I \|\nabla f(\mathbf{x})\| + \frac{1}{4}\epsilon^2 \eta I
$$

We can then use the inequality $x^2 \geq 2ax - a^2$ with $x = \|\nabla f(\mathbf{x})\|$ and $a = \epsilon$ to obtain

$$
V(\mathbf{x}) \leq -\frac{1}{4}\epsilon \eta I \|\nabla f(\mathbf{x})\| + \frac{1}{2}\epsilon^2 \eta I.
\tag{32}
$$

Having bounded $U(\mathbf{x})$ and $V(\mathbf{x})$, we can return to (25). Using (28), we can see

$$
U(\mathbf{x}) \leq -\frac{1}{2}\epsilon \eta I \|\nabla f(\mathbf{x})\| \leq -\frac{1}{4}\epsilon \eta I \|\nabla f(\mathbf{x})\| + \frac{1}{2}\epsilon^2 \eta I,
$$

so the RHS of (32) is an upper bound of both $U(\mathbf{x})$ and $V(\mathbf{x})$. Plugging this bound into (25) and taking total expectation then gives

$$
\mathbb{E}\left[ f(\bar{\boldsymbol{x}}_{r+1}) - f(\bar{\boldsymbol{x}}_r) \right] \leq -\frac{1}{4}\epsilon \eta I \mathbb{E}\left[ \|\nabla f(\bar{\boldsymbol{x}}_r)\| \right] + \frac{1}{2}\epsilon^2 \eta I.
$$

Finally, denoting $\Delta = f(\bar{\boldsymbol{x}}_0) - f^*$, we can unroll the above recurrence to obtain

$$
\mathbb{E}\left[ f(\bar{\boldsymbol{x}}_{R+1}) - f(\bar{\boldsymbol{x}}_0) \right] \leq -\frac{1}{4}\epsilon \eta I \sum_{r=0}^{R} \mathbb{E}\left[ \|\nabla f(\bar{\boldsymbol{x}}_r)\| \right] + \frac{1}{2}(R+1)\epsilon^2 \eta I,
$$

$$
\frac{1}{R+1} \sum_{r=0}^{R} \mathbb{E}\left[ \|\nabla f(\bar{\boldsymbol{x}}_r)\| \right] \leq \frac{4\Delta}{\epsilon \eta I (R+1)} + 2\epsilon,
$$

$$
\frac{1}{R+1} \sum_{r=0}^{R} \mathbb{E}\left[ \|\nabla f(\bar{\boldsymbol{x}}_r)\| \right] \leq 3\epsilon,
$$

where the last inequality comes from our choice of $R \geq \frac{4\Delta}{\epsilon^2 \eta I}$. $\qquad\square$

## D  DEFERRED PROOFS OF SECTION C

### D.1  PROOF OF CLAIM 1

*Proof.* Starting from Lemma 7 with $\boldsymbol{u} = \nabla f(\bar{\boldsymbol{x}}_r)$ and $\boldsymbol{v} = \boldsymbol{g}_t^i$, we have

$$
-\frac{\langle \nabla f(\bar{\boldsymbol{x}}_r), \boldsymbol{g}_t^i \rangle}{\|\boldsymbol{g}_t^i\|} \leq -\mu \|\nabla f(\bar{\boldsymbol{x}}_r)\| - (1-\mu)\|\boldsymbol{g}_t^i\| + (1+\mu)\|\boldsymbol{g}_t^i - \nabla f(\bar{\boldsymbol{x}}_r)\|.
\tag{33}
$$

Under $\bar{\mathcal{A}}_r = \{\|\boldsymbol{G}_r\| > \frac{\gamma}{\eta}\}$, note that $\boldsymbol{g}_t^i = \nabla F_i(\boldsymbol{x}_t^i; \xi_t^i) - \boldsymbol{G}_r^i + \boldsymbol{G}_r$, and we have

$$
\begin{aligned}
\|\boldsymbol{g}_t^i\| &\geq \|\boldsymbol{G}_r\| - \|\nabla F_i(\boldsymbol{x}_t^i, \xi_t^i) - \boldsymbol{G}_r^i\| \\
&\geq \frac{\gamma}{\eta} - \|\nabla F_i(\boldsymbol{x}_t^i, \xi_t^i) - \nabla f_i(\boldsymbol{x}_t^i)\| - \|\nabla f_i(\boldsymbol{x}_t^i) - \nabla f_i(\bar{\boldsymbol{x}}_r)\| - \|\nabla f_i(\bar{\boldsymbol{x}}_r) - \boldsymbol{G}_r^i\| \\
&\geq \frac{\gamma}{\eta} - 2\sigma - \|\nabla f_i(\boldsymbol{x}_t^i) - \nabla f_i(\bar{\boldsymbol{x}}_r)\|
\end{aligned}
$$

and

$$\|\boldsymbol{g}_t^i - \nabla f(\bar{\boldsymbol{x}}_r)\| \leq \|\nabla F_i(\boldsymbol{x}_t^i, \xi_t^i) - \nabla f_i(\boldsymbol{x}_t^i)\| + \|\nabla f_i(\boldsymbol{x}_t^i) - \nabla f_i(\bar{\boldsymbol{x}}_r)\|$$
$$+ \|\nabla f_i(\bar{\boldsymbol{x}}_r) - \boldsymbol{G}_r^i\| + \|\boldsymbol{G}_r - \nabla f(\bar{\boldsymbol{x}}_r)\|$$
$$\leq 3\sigma + \|\nabla f_i(\boldsymbol{x}_t^i) - \nabla f_i(\bar{\boldsymbol{x}}_r)\|.$$

Plugging these two inequalities into (33) yields

$$-\frac{\langle \nabla f(\bar{\boldsymbol{x}}_r), \boldsymbol{g}_t^i \rangle}{\|\boldsymbol{g}_t^i\|} \leq -\mu \|\nabla f(\bar{\boldsymbol{x}}_r)\| - (1-\mu)\frac{\gamma}{\eta} + (5+\mu)\sigma + 2\|\nabla f_i(\boldsymbol{x}_t^i) - \nabla f_i(\bar{\boldsymbol{x}}_r)\|.$$

Under $\bar{\mathcal{A}}_r$, we know $\|\boldsymbol{x}_t^i - \bar{\boldsymbol{x}}_r\| \leq \gamma I$, and $\gamma I \leq \frac{C}{L_1}$ by assumption. Therefore we can apply Lemma 6 to obtain

$$\|\nabla f_i(\boldsymbol{x}_t^i) - \nabla f_i(\bar{\boldsymbol{x}}_r)\| \leq (AL_0 + BL_1\|\nabla f_i(\bar{\boldsymbol{x}}_r)\|)\|\boldsymbol{x}_t^i - \bar{\boldsymbol{x}}_r\| \leq \gamma I(AL_0 + BL_1\|\nabla f_i(\bar{\boldsymbol{x}}_r)\|).$$

This implies that

$$-\frac{\langle \nabla f(\bar{\boldsymbol{x}}_r), \boldsymbol{g}_t^i \rangle}{\|\boldsymbol{g}_t^i\|} \leq -\mu \|\nabla f(\bar{\boldsymbol{x}}_r)\| - (1-\mu)\frac{\gamma}{\eta} + (5+\mu)\sigma + 2AL_0\gamma I + 2BL_1\gamma I\|\nabla f_i(\bar{\boldsymbol{x}}_r)\|.$$

Combining this with the choice $\mu = 2/5$, we have the final bound:

$$-\gamma\mathbb{E}_r\left[\frac{1}{N}\sum_{i=1}^{N}\sum_{t\in\mathcal{I}_r}\mathbb{1}(\bar{\mathcal{A}}_r)\langle \nabla f(\bar{\boldsymbol{x}}_r), \frac{\boldsymbol{g}_t^i}{\|\boldsymbol{g}_t^i\|}\rangle\right]$$

$$\leq \frac{1}{N}\sum_{i=1}^{N}(1-p_r)\left(-\frac{2}{5}\gamma I\|\nabla f(\bar{\boldsymbol{x}}_r)\| - \frac{3\gamma^2 I}{5\eta} + 6\gamma I\sigma + 2AL_0\gamma^2 I^2 + 2BL_1\gamma^2 I^2\|\nabla f_i(\bar{\boldsymbol{x}}_r)\|\right)$$

$$\leq (1-p_r)\left(\left(-\frac{2}{5}\gamma I + 2BL_1\rho\gamma^2 I^2\right)\|\nabla f(\bar{\boldsymbol{x}}_r)\| - \frac{3\gamma^2 I}{5\eta} + 2\gamma^2 I^2(AL_0 + BL_1\kappa) + 6\gamma I\sigma\right)$$

where we used the heterogeneity assumption $\|\nabla f_i(\bar{\boldsymbol{x}}_r)\| \leq \kappa + \rho\|\nabla f(\bar{\boldsymbol{x}}_r)\|$. $\qquad\square$

## D.2   PROOF OF CLAIM 2

*Proof.* Recall the event $\mathcal{A}_r = \{\|\boldsymbol{G}_r\| \leq \gamma/\eta\}$, we have

$$I\mathbb{E}_r\left[\frac{1}{N}\sum_{i=1}^{N}\sum_{t\in\mathcal{I}_r}\mathbb{1}(\mathcal{A}_r)\langle \nabla f(\bar{\boldsymbol{x}}_r), \boldsymbol{g}_t^i\rangle\right] = \mathbb{E}_r\left[\mathbb{1}(\mathcal{A}_r)\left\langle I\nabla f(\bar{\boldsymbol{x}}_r), \sum_{t\in\mathcal{I}_r}\frac{1}{N}\sum_{i=1}^{N}\boldsymbol{g}_t^i\right\rangle\right]$$

$$\overset{(i)}{=} \mathbb{E}_r\left[\mathbb{1}(\mathcal{A}_r)\left\langle I\nabla f(\bar{\boldsymbol{x}}_r), \sum_{t\in\mathcal{I}_r}\frac{1}{N}\sum_{i=1}^{N}\nabla F_i(\boldsymbol{x}_t^i; \xi_t^i)\right\rangle\right]$$

$$\overset{(ii)}{=} \mathbb{E}_r\left[\mathbb{1}(\mathcal{A}_r)\left\langle I\nabla f(\bar{\boldsymbol{x}}_r), \sum_{t\in\mathcal{I}_r}\frac{1}{N}\sum_{i=1}^{N}\nabla f_i(\boldsymbol{x}_t^i)\right\rangle\right]$$

$$\overset{(iii)}{=} \frac{p_r I^2}{2}\|\nabla f(\bar{\boldsymbol{x}}_r)\|^2 + \frac{1}{2}\mathbb{E}_r\left[\mathbb{1}(\mathcal{A}_r)\left\|\frac{1}{N}\sum_{i=1}^{N}\sum_{t\in\mathcal{I}_r}\nabla f(\boldsymbol{x}_t^i)\right\|^2\right]$$

$$-\frac{1}{2}\mathbb{E}_r\left[\mathbb{1}(\mathcal{A}_r)\left\|\sum_{t\in\mathcal{I}_r}\left(\frac{1}{N}\sum_{i=1}^{N}\nabla f_i(\boldsymbol{x}_t^i) - \nabla f(\bar{\boldsymbol{x}}_r)\right)\right\|^2\right]. \tag{34}$$

The equality $(i)$ is obtained from the fact that $\frac{1}{N}\sum_{i=1}^{N}\boldsymbol{g}_t^i = \frac{1}{N}\sum_{i=1}^{N}\nabla F_i(\boldsymbol{x}_t^i, \xi_t^i) - \boldsymbol{G}_r^i + \boldsymbol{G}_r = \frac{1}{N}\sum_{i=1}^{N}\nabla F_i(\boldsymbol{x}_t^i, \xi_t^i)$. The equality $(ii)$ holds due to the tower property such that for $t > t_r$

$$\mathbb{E}\left[\mathbb{1}(\mathcal{A}_r)\nabla F_i(\boldsymbol{x}_t^i, \xi_t^i)\big|\mathcal{F}_r\right] = \mathbb{E}\left[\mathbb{1}(\mathcal{A}_r)\mathbb{E}\left[\nabla F_i(\boldsymbol{x}_t^i, \xi_t^i)\big|\mathcal{H}_t\right]\big|\mathcal{F}_r\right] = \mathbb{E}\left[\mathbb{1}(\mathcal{A}_r)\nabla f_i(\boldsymbol{x}_t^i)\big|\mathcal{F}_r\right];$$

for $t = t_r$

$$\mathbb{E}\left[\mathbb{1}(\mathcal{A}_r)\nabla F_i(\bar{\boldsymbol{x}}_r, \xi^i_{t_r})\big|\mathcal{F}_r\right] = \mathbb{E}\left[\mathbb{1}(\mathcal{A}_r)|\mathcal{F}_r\right]\mathbb{E}\left[\nabla F_i(\bar{\boldsymbol{x}}_r, \xi^i_{t_r})\big|\mathcal{F}_r\right] = \mathbb{E}\left[\mathbb{1}(\mathcal{A}_r)\nabla f_i(\bar{\boldsymbol{x}}_r)\big|\mathcal{F}_r\right],$$

which is true since $\boldsymbol{G}_r = \frac{1}{N}\sum_{i=1}^N \nabla F_i(\bar{\boldsymbol{x}}_r; \widetilde{\xi}^i_r)$ is independent of $\nabla F_i(\bar{\boldsymbol{x}}_r, \xi^i_{t_r})$ given $\mathcal{F}_r$, and $(iii)$ holds because $2\langle a, b\rangle = \|a\|^2 + \|b\|^2 - \|a - b\|^2$.

Let $\Gamma = AL_0 + BL_1\left(\kappa + \rho\left(\sigma + \frac{\gamma}{\eta}\right)\right)$. Notice that we can apply the relaxed smoothness in Lemma 6 to obtain

$$\mathbb{E}_r\left[\mathbb{1}(\mathcal{A}_r)\|\nabla f_i(\boldsymbol{x}^i_t) - \nabla f_i(\bar{\boldsymbol{x}}_r)\|^2\right]$$
$$\leq \mathbb{E}_r\left[\mathbb{1}(\mathcal{A}_r)(AL_0 + BL_1\|\nabla f_i(\bar{\boldsymbol{x}}_r)\|)^2\|\boldsymbol{x}^i_t - \bar{\boldsymbol{x}}_r\|^2\right]$$
$$\leq \mathbb{E}_r\left[\mathbb{1}(\mathcal{A}_r)(AL_0 + BL_1(\kappa + \rho\|\nabla f(\bar{\boldsymbol{x}}_r)\|))^2\|\boldsymbol{x}^i_t - \bar{\boldsymbol{x}}_r\|^2\right]$$
$$\overset{(i)}{\leq} \Gamma^2\mathbb{E}_r\left[\mathbb{1}(\mathcal{A}_r)\|\boldsymbol{x}^i_t - \bar{\boldsymbol{x}}_r\|^2\right]$$
$$\overset{(ii)}{\leq} 18p_r I^2\eta^2\Gamma^2\left(2\|\nabla f(\bar{\boldsymbol{x}}_r)\|^2 + 7\sigma^2\right).$$

The inequality $(i)$ holds since $\|\nabla f(\bar{\boldsymbol{x}}_r)\| \leq \|\nabla f(\bar{\boldsymbol{x}}_r) - \boldsymbol{G}_r\| + \|\boldsymbol{G}_r\| \leq \sigma + \gamma/\eta$ almost surely under the event $\mathcal{A}_r$. The inequality $(ii)$ follows from the bound (14) in Lemma 3. Therefore, we are guaranteed that

$$\mathbb{E}_r\left[\mathbb{1}(\mathcal{A}_r)\left\|\sum_{t\in\mathcal{I}_r}\left(\frac{1}{N}\sum_{i=1}^N \nabla f_i(\boldsymbol{x}^i_t) - \nabla f(\bar{\boldsymbol{x}}_r)\right)\right\|^2\right]$$
$$\leq I\sum_{t\in\mathcal{I}_r}\frac{1}{N}\sum_{i=1}^N \mathbb{E}_r\left[\mathbb{1}(\mathcal{A}_r)\left\|\nabla f_i(\boldsymbol{x}^i_t) - \nabla f(\bar{\boldsymbol{x}}_r)\right\|^2\right]$$
$$\leq I\sum_{t\in\mathcal{I}_r}\frac{1}{N}\sum_{i=1}^N 18p_r I^2\eta^2\Gamma^2\left(2\|\nabla f(\bar{\boldsymbol{x}}_r)\|^2 + 7\sigma^2\right)$$
$$\leq 18p_r I^4\eta^2\Gamma^2\left(2\|\nabla f(\bar{\boldsymbol{x}}_r)\|^2 + 7\sigma^2\right). \tag{35}$$

Multiplying both sides of (34) by $-\eta/I$ and substituting (35) then yields

$$-\eta\mathbb{E}_r\left[\frac{1}{N}\sum_{i=1}^N\sum_{t\in\mathcal{I}_r}\mathbb{1}(\mathcal{A}_r)\langle\nabla f(\bar{\boldsymbol{x}}_r), \boldsymbol{g}^i_t\rangle\right]$$
$$\leq -\frac{p_r\eta I}{2}\|\nabla f(\bar{\boldsymbol{x}}_r)\|^2 - \frac{\eta}{2I}\mathbb{E}_r\left[\mathbb{1}(\mathcal{A}_r)\left\|\frac{1}{N}\sum_{i=1}^N\sum_{t\in\mathcal{I}_r}\nabla f(\boldsymbol{x}^i_t)\right\|^2\right]$$
$$+ \frac{p_r\eta}{2I}\mathbb{E}_r\left[\left\|\sum_{t\in\mathcal{I}_r}\left(\frac{1}{N}\sum_{i=1}^N \nabla f_i(\boldsymbol{x}^i_t) - \nabla f(\bar{\boldsymbol{x}}_r)\right)\right\|^2\right]$$
$$\leq p_r\left[\left(-\frac{\eta I}{2} + 36I^3\eta^3\Gamma^2\right)\|\nabla f(\bar{\boldsymbol{x}}_r)\|^2 + 126I^3\eta^3\sigma^2\Gamma^2\right] - \frac{\eta}{2I}\mathbb{E}_r\left[\mathbb{1}(\mathcal{A}_r)\left\|\frac{1}{N}\sum_{i=1}^N\sum_{t\in\mathcal{I}_r}\nabla f(\boldsymbol{x}^i_t)\right\|^2\right].$$

$\square$

### D.3 PROOF OF CLAIM 3

*Proof.* From the definition of $\bar{\boldsymbol{x}}_{r+1}$, we have

$$\mathbb{E}_r\left[\|\bar{\boldsymbol{x}}_{r+1} - \bar{\boldsymbol{x}}_r\|^2\right]$$

$$\leq 2\eta^2 \mathbb{E}_r\left[\mathbb{1}(\mathcal{A}_r)\left\|\frac{1}{N}\sum_{i=1}^N \sum_{t\in\mathcal{I}_r} \boldsymbol{g}_t^i\right\|^2\right] + 2\gamma^2 \mathbb{E}_r\left[\mathbb{1}(\bar{\mathcal{A}}_r)\left\|\frac{1}{N}\sum_{i=1}^N \sum_{t\in\mathcal{I}_r} \frac{\boldsymbol{g}_t^i}{\|\boldsymbol{g}_t^i\|}\right\|^2\right]$$

$$\overset{(i)}{\leq} 2\eta^2 \mathbb{E}_r\left[\mathbb{1}(\mathcal{A}_r)\left\|\frac{1}{N}\sum_{i=1}^N \sum_{t\in\mathcal{I}_r} \nabla F_i(\boldsymbol{x}_t^i, \xi_t^i)\right\|^2\right] + 2(1-p_r)\gamma^2 I^2$$

$$\leq 4\eta^2 \mathbb{E}_r\left[\mathbb{1}(\mathcal{A}_r)\left\|\frac{1}{N}\sum_{i=1}^N \sum_{t\in\mathcal{I}_r} \nabla f_i(\boldsymbol{x}_t^i)\right\|^2\right]$$

$$+ 4p_r\eta^2 \mathbb{E}_r\left[\mathbb{1}(\mathcal{A}_r)\left\|\frac{1}{N}\sum_{i=1}^N \sum_{t\in\mathcal{I}_r} \nabla F_i(\boldsymbol{x}_t^i; \xi_t^i) - \nabla f_i(\boldsymbol{x}_t^i)\right\|^2\right] + 2(1-p_r)\gamma^2 I^2$$

$$\overset{(ii)}{\leq} 4\eta^2 \mathbb{E}_r\left[\mathbb{1}(\mathcal{A}_r)\left\|\frac{1}{N}\sum_{i=1}^N \sum_{t\in\mathcal{I}_r} \nabla f_i(\boldsymbol{x}_t^i)\right\|^2\right]$$

$$+ 4\eta^2 \frac{1}{N^2}\sum_{i=1}^N \mathbb{E}_r\left[\mathbb{1}(\mathcal{A}_r)\left\|\sum_{t\in\mathcal{I}_r} \nabla F_i(\boldsymbol{x}_t^i; \xi_t^i) - \nabla f_i(\boldsymbol{x}_t^i)\right\|^2\right] + 2(1-p_r)\gamma^2 I^2, \quad (36)$$

where $(i)$ is obtained by noticing that $\frac{1}{N}\sum_{i=1}^N \boldsymbol{g}_t^i = \frac{1}{N}\sum_{i=1}^N \nabla F_i(\boldsymbol{x}_t^i, \xi_t^i)$, and $(ii)$ holds by the fact that each client's stochastic gradients $\nabla F_i(\boldsymbol{x}_t^i, \xi_t^i)$ are sampled independently from one another. Similarly, let $s \in \mathcal{I}_r$ with $s > t$., we can see that

$$\mathbb{E}_r\left[\mathbb{1}(\mathcal{A}_r)\langle \nabla F_i(\boldsymbol{x}_t^i; \xi_t^i) - \nabla f_i(\boldsymbol{x}_t^i), \nabla F_i(\boldsymbol{x}_s^i; \xi_s^i) - \nabla f_i(\boldsymbol{x}_s^i)\rangle\right]$$

$$= \mathbb{E}_r\left[\mathbb{1}(\mathcal{A}_r)\mathbb{E}_r\left[\langle \nabla F_i(\boldsymbol{x}_t^i; \xi_t^i) - \nabla f_i(\boldsymbol{x}_t^i), \nabla F_i(\boldsymbol{x}_s^i; \xi_s^i) - \nabla f_i(\boldsymbol{x}_s^i)\rangle\Big|\mathcal{H}_s\right]\right]$$

$$= \mathbb{E}_r\left[\mathbb{1}(\mathcal{A}_r)\langle \nabla F_i(\boldsymbol{x}_t^i; \xi_t^i) - \nabla f_i(\boldsymbol{x}_t^i), \mathbb{E}_r\left[\nabla F_i(\boldsymbol{x}_s^i; \xi_s^i)\Big|\mathcal{H}_s\right] - \nabla f_i(\boldsymbol{x}_s^i)\rangle\right]$$

$$= 0.$$

Therefore, we have

$$\frac{1}{N^2}\sum_{i=1}^N \mathbb{E}_r\left[\mathbb{1}(\mathcal{A}_r)\left\|\sum_{t\in\mathcal{I}_r} \nabla F_i(\boldsymbol{x}_t^i; \xi_t^i) - \nabla f_i(\boldsymbol{x}_t^i)\right\|^2\right]$$

$$= \frac{1}{N^2}\sum_{i=1}^N \sum_{t\in\mathcal{I}_r} \mathbb{E}_r\left[\mathbb{1}(\mathcal{A}_r)\left\|\nabla F_i(\boldsymbol{x}_t^i; \xi_t^i) - \nabla f_i(\boldsymbol{x}_t^i)\right\|^2\right]$$

$$= \frac{1}{N^2}\sum_{i=1}^N \sum_{t\in\mathcal{I}_r} \mathbb{E}_r\left[\mathbb{1}(\mathcal{A}_r)\mathbb{E}_r\left[\left\|\nabla F_i(\boldsymbol{x}_t^i; \xi_t^i) - \nabla f_i(\boldsymbol{x}_t^i)\right\|^2\Big|\mathcal{H}_t\right]\right]$$

$$\leq \frac{p_r I \sigma^2}{N}. \quad (37)$$

And the desired result is obtained by plugging (37) into (36). $\qquad\square$

## E ADDITIONAL EXPERIMENTAL RESULTS

### E.1    PROOF OF PROPOSITION 1

*Proof.* Recall the definition of $f_1(x)$ and $f_2(x)$,

$$f_1(x) = x^4 - 3x^3 + Hx^2 + x, \quad f_2(x) = x^4 - 3x^3 - 2Hx^2 + x,$$

which means

$$\nabla f(x) = 4x^3 - 9x^2 - Hx + 1.$$

and

$$\nabla f_1(x) - \nabla f(x) = 3Hx, \quad \nabla f_2(x) - \nabla f(x) = -3Hx,$$

It follows that

$$
\begin{aligned}
\|\nabla f_i(x)\| &\le \|\nabla f_i(x) - \nabla f(x)\| + \|\nabla f(x)\| \\
&\le 3H|x| + \|\nabla f(x)\| \\
&\le 3H|x| - \left|4x^3 - 9x^2 - Hx + 1\right| + 2\|\nabla f(x)\| \\
&\le 4H|x| - \left|4x^3 - 9x^2 + 1\right| + 2\|\nabla f(x)\| \\
&\le 10H|x| - \left|4x^3 - 9x^2\right| + 1 + 2\|\nabla f(x)\|.
\end{aligned}
\tag{38}
$$

Let $g(x) = 10H|x| - \left|4x^3 - 9x^2\right|$, next we will characterize $g(x)$ in different region.

(i) When $x \in (-\infty, 0)$, $g(x) = 4x^3 - 9x^2 - 10Hx$. The root for the derivative of $g(x)$ in this region is

$$12x^2 - 18x - 10H = 0 \Longrightarrow x = x_1 := \frac{18 - \sqrt{18^2 + 480H}}{24}.$$

It follows that

$$
\begin{aligned}
g(x) &\le 4x_1^3 - 9x_1^2 - 10Hx_1 \\
&\le 10H\left(\frac{\sqrt{18^2 + 480H} - 18}{24}\right) \\
&\le 10H\left(\frac{20H}{24}\right) \le \frac{25H^2}{3}.
\end{aligned}
\tag{39}
$$

where the last inequality follows from $x_1 \le 0$.

(ii) When $x \in (0, \frac{9}{4})$, $g(x) = 4x^3 - 9x^2 + 10Hx$. The derivative of $g(x)$ is greater than 0 in this case since $18^2 - 480H \le 0$ for $H \ge 1$. Then we have

$$g(x) \le 10H \cdot \frac{9}{4} = \frac{45H}{2}.
\tag{40}$$

(iii) When $x \in (\frac{9}{4}, +\infty)$, $g(x) = -4x^3 + 9x^2 + 10Hx$. The root for the derivative of $g(x)$ is

$$-12x^2 + 18x + 10H = 0 \Longrightarrow x = x_2 := \frac{-18 + \sqrt{18^2 + 480H}}{24}.$$

Then we have

$$
\begin{aligned}
g(x) &\le \max\left\{-4x_2^3 + 9x_2^2 + 10Hx_2, -4\left(\frac{9}{4}\right)^3 + 9\left(\frac{9}{4}\right)^2 + \frac{45H}{2}\right\} \\
&\le 9x_2^2 + 10Hx_2 + 9\left(\frac{9}{4}\right)^2 + \frac{45H}{2}.
\end{aligned}
\tag{41}
$$

Combining (39), (40) and (41), we are guaranteed that

$$
\begin{aligned}
g(x) + 1 &\le 9\left(\frac{-18 + \sqrt{18^2 + 480H}}{24}\right)^2 + 10H\left(\frac{-18 + \sqrt{18^2 + 480H}}{24}\right) \\
&\quad + \frac{25H^2}{3} + 45H + 100 \\
&:= \kappa(H).
\end{aligned}
$$

Substituting this bound into (38), we get

$$\|\nabla f_i(x)\| \leq 2\|\nabla f(x)\| + g(x) + 1$$
$$\leq 2\|\nabla f(x)\| + \kappa(H).$$

And $\kappa(H) < \infty$ is an increasing function of $H$. □

### E.2 SYNTHETIC TASK

For two algorithms, we inject uniform noise over $[-1, 1]$ into the gradient at each step, and tune $\gamma/\eta \in \{5, 10, 15\}$ and tune $\eta \in \{0.1, 0.01, 0.001\}$. We run each algorithm for 500 communication rounds and the length of each communication round is $I = 8$. The results are showed in Figure 3.

### E.3 SNLI

The learning rate $\eta$ and the clipping parameter $\gamma$ are tuned with search in the following way: we vary $\gamma \in \{0.01, 0.03, 0.1\}$ and for each $\gamma$ we vary $\eta$ so that the clipping threshold $\gamma/\eta$ varies over $\{0.1, 0.333, 1.0, 3.333, 10.0\}$, leading to 15 pairs $(\eta, \gamma)$. We decay both $\eta$ and $\gamma$ by a factor of $0.5$ at epochs 15 and 20. We choose the best pair $(\eta, \gamma)$ according to the performance on a validation set, and the corresponding model is evaluated on a held-out test set. Note that we do not tune $(\gamma, \eta)$ separately for each algorithm. Instead, due to computational constraints, we tune the hyperparameters for the baseline CELGC under the setting $I = 4$, $s = 50\%$ and re-use the tuned values for the rest of the settings.

### E.4 CIFAR-10

#### E.4.1 SETUP

We train a ResNet-50 (He et al., 2016) for 150 epochs using the cross-entropy loss and a batch size of 64 for each worker. Starting from an initial learning rate $\eta_0 = 1.0$ and clipping parameter $\gamma = 0.5$, we decay the learning rate by a factor of $0.5$ at epochs 80 and 120. In this setting, we decay the clipping parameter $\gamma$ with the learning rate $\eta$, so that the clipping threshold $\frac{\gamma}{\eta}$ remains constant during training. We present results for $I = 8$ and $s \in \{50\%, 70\%\}$. We include the same baselines as the experiments of the main text, comparing EPISODE to FedAvg, SCAFFOLD, and CELGC.

#### E.4.2 RESULTS

Training loss and testing accuracy during training are shown below in Figure 4. In both settings, EPISODE is superior in terms of testing accuracy and nearly the best in terms of training loss.

### E.5 IMAGENET

The training curves (training and testing loss) for each ImageNet setting are shown below in Figure 5.

## F RUNNING TIME RESULTS

To demonstrate the utility of EPISODE for federated learning in practical settings, we also provide a comparison of the running time of each algorithm on the SNLI dataset. Our experiments were run on eight NVIDIA Tesla V100 GPUs distributed on two machines. The training loss and testing accuracy of each algorithm (under the settings described above) are plotted against running time below. Note that these are the same results as shown in Figure 1, plotted against time instead of epochs or communication rounds.

On the SNLI dataset, EPISODE reaches a lower training loss and higher testing accuracy with respect to time, compared with CELGC and NaiveParallelClip. Table 2 shows that, when $I \leq 8$, EPISODE requires significantly less running time to reach high testing accuracy compared with both CELGC and NaiveParallelClip. When $I = 16$, CELGC and NaiveParallelClip nearly match, indicating that $I = 16$ may be close to the theoretical upper bound on $I$ for which fast convergence can be guaranteed. Also, as the client data similarity decreases, the running time requirement of EPISODE

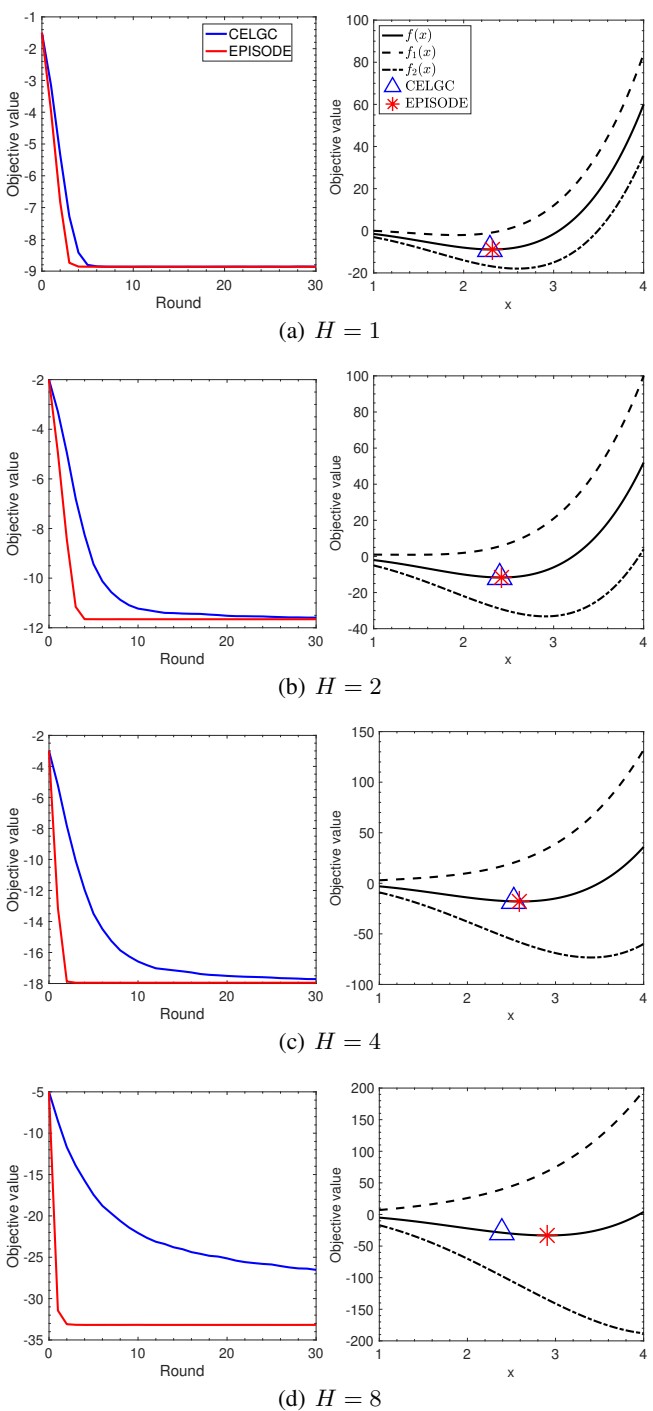

Figure 3: The loss trajectories and converged solutions of CELGC and EPISODE on synthetic task.

to reach high test accuracy stays nearly constant (e.g., when $I = 4$), while the running time required by CELGC steadily increases. This demonstrates the resilience of EPISODE's convergence speed to heterogeneity. Training curves for the same experiment are shown in Figure 6.

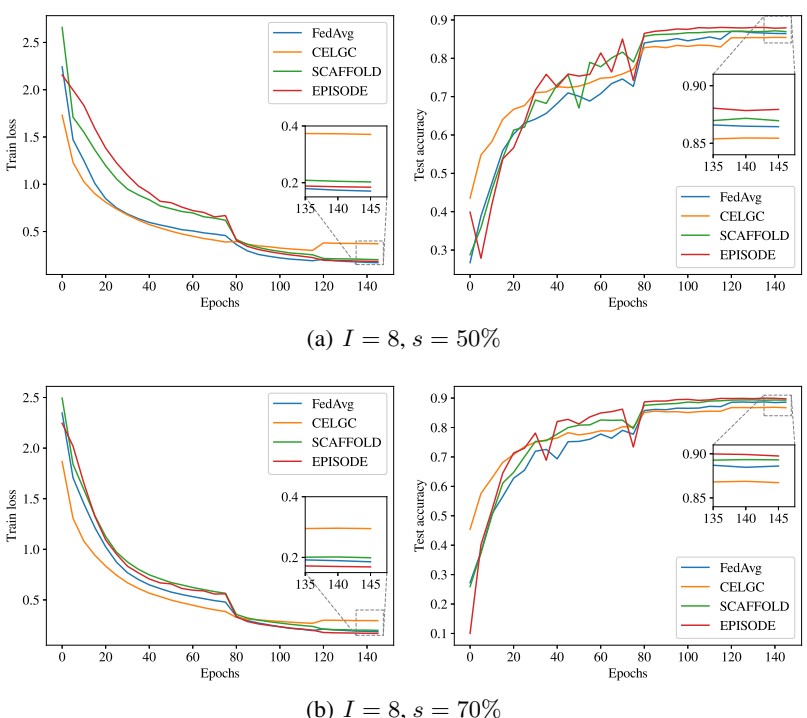

(a) $I = 8$, $s = 50\%$

(b) $I = 8$, $s = 70\%$

Figure 4: Training curves for CIFAR-10 experiments.

| Interval | Similarity | Algorithm | 70% | 75% | 80% |
|---|---|---|---|---|---|
| 1 | 100% | NaiveParallelClip | 37.30 | 59.69 | 118.45 |
| 2 | 30% | CELGC | 33.57 | 63.98 | N/A |
| | | EPISODE | **27.20** | **38.07** | **70.60** |
| 4 | 30% | CELGC | 23.84 | 42.51 | N/A |
| | | EPISODE | **18.34** | **25.73** | **55.15** |
| 8 | 30% | CELGC | 20.37 | 34.06 | N/A |
| | | EPISODE | **13.98** | **22.43** | **53.43** |
| 16 | 30% | CELGC | **16.57** | **27.00** | N/A |
| | | EPISODE | 21.26 | 28.39 | N/A |
| 4 | 50% | CELGC | 18.52 | 31.86 | N/A |
| | | EPISODE | **18.37** | **25.71** | **47.76** |
| 4 | 10% | CELGC | 39.75 | N/A | N/A |
| | | EPISODE | **18.46** | **29.71** | **55.92** |

Table 2: Running time (in minutes) for each algorithm to reach test accuracy of $70\%$, $75\%$, and $80\%$ on SNLI dataset. We use N/A to denote when an algorithm did not reach the corresponding level of accuracy over the course of training.

## G  ABLATION STUDY

In this section, we introduce an ablation study which disentangles the role of the two components of EPISODE's algorithm design: periodic resampled corrections and episodic clipping. Using the SNLI dataset, we have evaluated several variants of the EPISODE algorithm constructed by removing one algorithmic component at a time, and we compare the performance against EPISODE along with variants of the baselines mentioned in the paper. Our ablation study shows that both components of

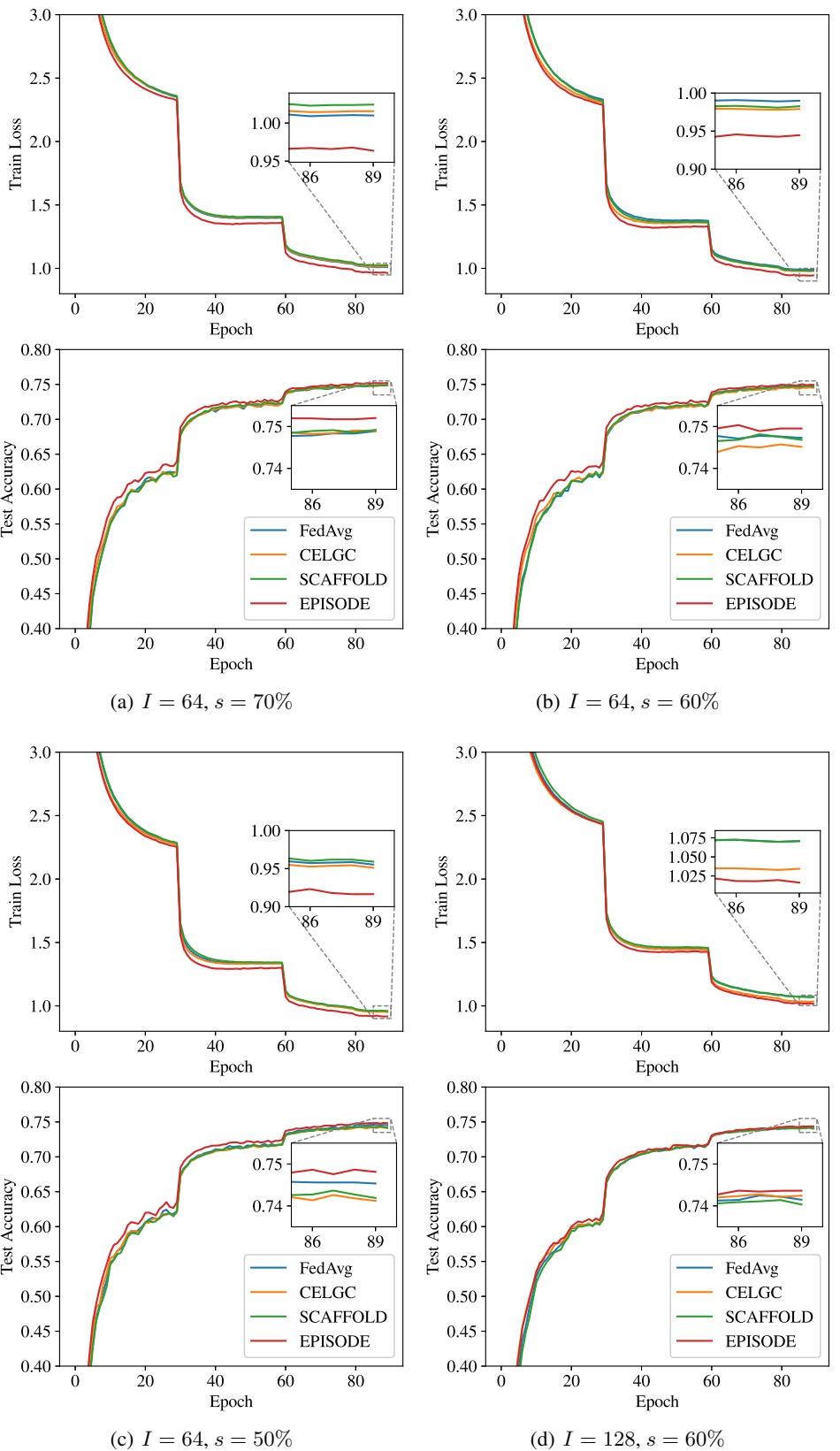

Figure 5: Training curves for all ImageNet experiments.

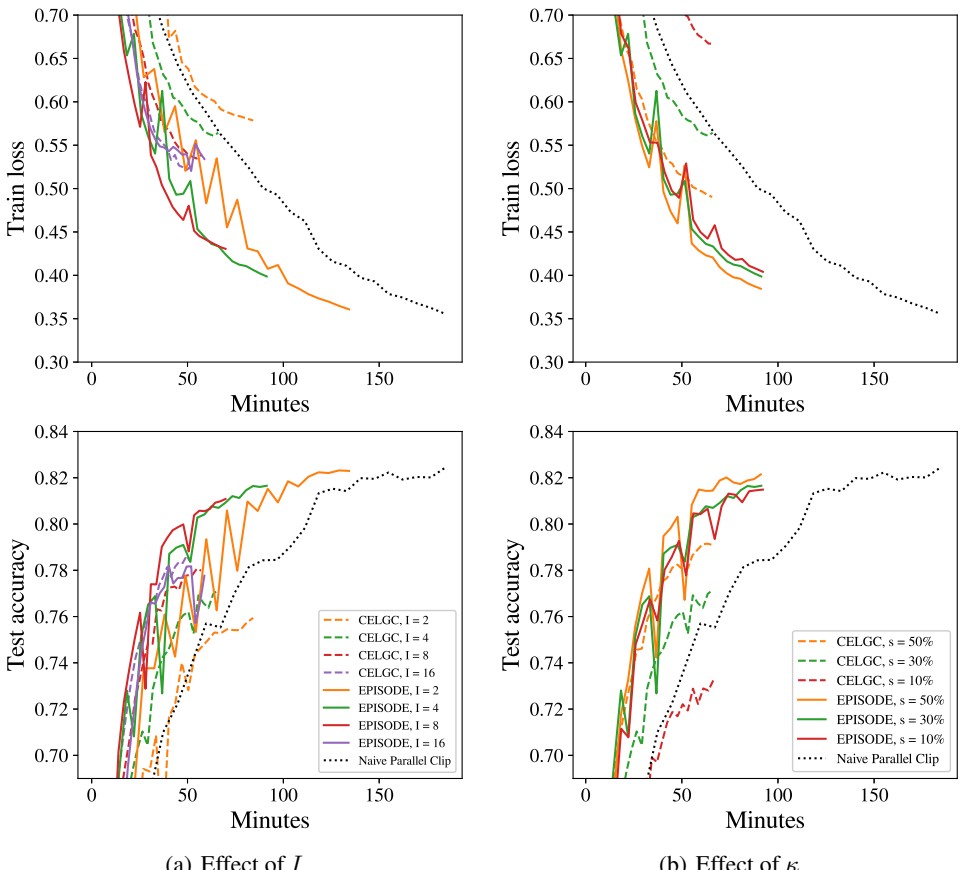

(a) Effect of $I$  (b) Effect of $\kappa$

Figure 6: Training loss and testing accuracy on SNLI against running time. **(a)** Various values of communication intervals $I \in \{2, 4, 8, 16\}$ with fixed data similarity $s = 30\%$. **(b)** Various values of data similarity $s \in \{10\%, 30\%, 50\%\}$ with fixed $I = 4$.

EPISODE's algorithm design (periodically resampled corrections and episodic clipping) contribute to the improved performance over previous work.

Our ablation experiments follow the same setting as the SNLI experiments in the main text. The network architecture, hyperparameters, and dataset are all identical to the SNLI experiments described in the main text. In this ablation study, we additionally evaluate multiple variants of EPISODE and baselines, which are described below:

- SCAFFOLD (clipped): The SCAFFOLD algorithm (Karimireddy et al., 2020) with gradient clipping applied at each iteration. This algorithm, as a variant of CELGC, determines the gradient clipping operation based on the corrected gradient at every iteration on each machine.

- EPISODE (unclipped): The EPISODE algorithm with clipping operation removed.

- FedAvg: The FedAvg algorithm (McMahan et al., 2017a). We include this to show that clipping in some form is crucial for optimization in the relaxed smoothness setting.

- SCAFFOLD: The SCAFFOLD algorithm (Karimireddy et al., 2020). We include this to show that SCAFFOLD-style corrections are not sufficient for optimization in the relaxed smoothness setting.

We compare these four algorithm variations against the algorithms discussed in the main text, which include EPISODE, CELGC, and NaiveParallelClip.

| Interval | Similarity | Algorithm | Train Loss | Test Acc. |
|---|---|---|---|---|
| 1 | 100% | NaiveParallelClip | 0.357 | 82.4% |
| 2 | 30% | CELGC | 0.579 | 75.9% |
| | | EPISODE | **0.361** | **82.3%** |
| | | SCAFFOLD (clipped) | 0.445 | 80.5% |
| | | EPISODE (unclipped) | 4.51 | 33.3% |
| | | FedAvg | 1.56 | 32.8% |
| | | SCAFFOLD | 1.23 | 34.1% |
| 4 | 30% | CELGC | 0.564 | 77.2% |
| | | EPISODE | **0.399** | **81.7%** |
| | | SCAFFOLD (clipped) | 0.440 | 80.7% |
| | | EPISODE (unclipped) | 9.82 | 33.0% |
| | | FedAvg | 1.14 | 32.8% |
| | | SCAFFOLD | 4.39 | 32.8% |
| 8 | 30% | CELGC | 0.539 | 78.0% |
| | | EPISODE | **0.431** | **81.1%** |
| | | SCAFFOLD (clipped) | 0.512 | 77.1% |
| | | EPISODE (unclipped) | 8.02 | 34.3% |
| | | FedAvg | 1.25 | 32.7% |
| | | SCAFFOLD | 10.86 | 32.8% |
| 16 | 30% | CELGC | 0.525 | 78.3% |
| | | EPISODE | **0.534** | **77.8%** |
| | | SCAFFOLD (clipped) | 0.597 | 75.7% |
| | | EPISODE (unclipped) | 4.71 | 33.0% |
| | | FedAvg | 3.45 | 32.7% |
| | | SCAFFOLD | 4.87 | 32.7% |
| 4 | 50% | CELGC | 0.490 | 79.1% |
| | | EPISODE | **0.385** | **82.1%** |
| | | SCAFFOLD (clipped) | 0.436 | 80.7% |
| | | EPISODE (unclipped) | 9.08 | 34.3% |
| | | FedAvg | 4.81 | 32.8% |
| | | SCAFFOLD | 2.40 | 32.9% |
| 4 | 10% | CELGC | 0.667 | 73.3% |
| | | EPISODE | **0.404** | **81.5%** |
| | | SCAFFOLD (clipped) | 0.438 | 80.7% |
| | | EPISODE (unclipped) | 8.54 | 33.0% |
| | | FedAvg | 1.89 | 34.3% |
| | | SCAFFOLD | 5.61 | 34.3% |

Table 3: Results for ablation study of EPISODE on SNLI dataset.

Following the protocol outlined in the main text, we train each one of these algorithms while varying the communication interval $I$ and the client data similarity parameter $s$. Specifically, we evaluate six settings formed by first fixing $s = 30\%$ and varying $I \in \{2, 4, 8, 16\}$, then fixing $I = 4$ and varying $s \in \{10\%, 30\%, 50\%\}$. Note that the results of NaiveParallelClip are unaffected by $I$ and $s$, since NaiveParallelClip communicates at every iteration. For each of these six settings, we provide the training loss and testing accuracy reached by each algorithm at the end of training. Final results for all settings are given in Table 3, and training curves for the setting $I = 4, s = 30\%$ are shown in Figure 7.

From these results, we can conclude that both components of EPISODE (periodic resampled corrections and episodic clipping) contribute to EPISODE's improved performance.

- Replacing periodic resampled corrections with SCAFFOLD-style corrections yields the variant SCAFFOLD (clipped). In every setting, SCAFFOLD (clipped) performs slightly

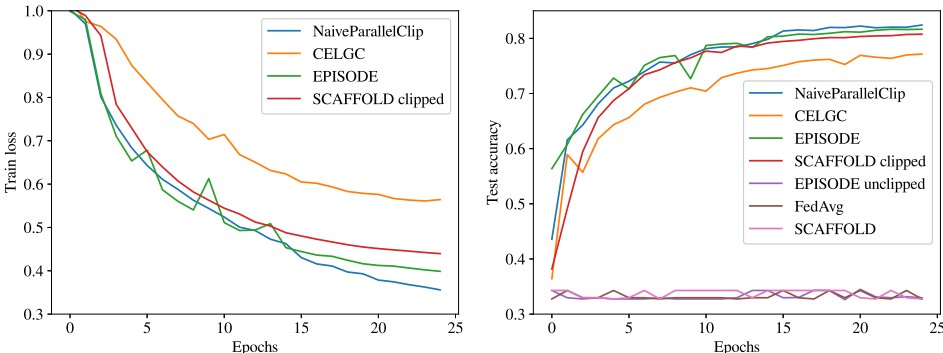

Figure 7: Training curves SNLI ablation study under the setting $I = 4$ and $s = 30\%$. Note that the training losses of EPISODE (unclipped), FedAvg, and SCAFFOLD are not visible, since they are orders of magnitude larger than the other algorithms.

better than CELGC, but still worse than EPISODE. This corroborates the intuition that SCAFFOLD-style corrections use slightly outdated information compared to that of EPISODE, and this information lag caused worse performance in this ablation study.

- On the other hand, clipping is essential for EPISODE to avoid divergence. By removing clipping from EPISODE, we obtain the variant EPISODE (unclipped), which fails to learn entirely. EPISODE (unclipped) never reached a test accuracy higher than $35\%$, which is barely higher than random guessing, since SNLI is a 3-way classification problem. In summary, both periodic resampled corrections and episodic clipping contribute to the improved performance of EPISODE over baselines.

In addition, FedAvg and SCAFFOLD show similar *divergence* behavior as EPISODE (unclipped). None of these three algorithms employ any clipping or normalization in updates, and consequently none of these algorithms are able to surpass random performance on SNLI. Finally, although NaiveParallelClip appears to be the best performing algorithm from this table, it requires more wall-clock time than any other algorithms due to its frequent communication. For a comparison of the running time results, see Table 2 in Appendix F.

## H    NEW EXPERIMENTS ON FEDERATED LEARNING BENCHMARK: SENTIMENT140 DATASET

To evaluate EPISODE on a real-world federated dataset, we provide additional experiments on the Sentiment140 benchmark from the LEAF benchmark (Caldas et al., 2018). Sentiment140 is a sentiment classification problem on a dataset of tweets, where each tweet is labeled as positive or negative. For this setting, we follow the experimental setup of Li et al. (2020b): training a 2-layer LSTM network with 256 hidden units on the cross-entropy classification loss. We also follow their data preprocessing steps to eliminate users with a small number of data points and split into training and testing sets. We perform an additional step to simulate the cross-silo federated environment (Kairouz et al., 2019) by partitioning the original Sentiment140 users into eight groups (i.e., eight machines). To simulate heterogeneity between silos, we partition the users based on a non-i.i.d. sampling scheme similar to that of our SNLI experiments. Specifically, given a silo similarity parameter $s$, each silo is allocated $s\%$ of its users by uniform sampling, and $(100 - s)\%$ of its users from a pool of users which are sorted by the proportion of positive tweets in their local dataset. This way, when $s$ is small, different silos will have a very different proportion of positive/negative samples in their respective datasets. We evaluate NaiveParallelClip, CELGC, and EPISODE in this cross-silo environment with $I = 4$ and $s \in \{0, 10, 20\}$. We tuned the learning rate $\eta$, and the clipping parameter $\gamma$ with grid search over the values $\eta \in \{0.01, 0.03, 0.1, 0.3, 1.0\}$ and $\gamma \in \{0.01, 0.03, 0.1, 0.3, 1.0\}$. Results are plotted in Figures 8 and 9.

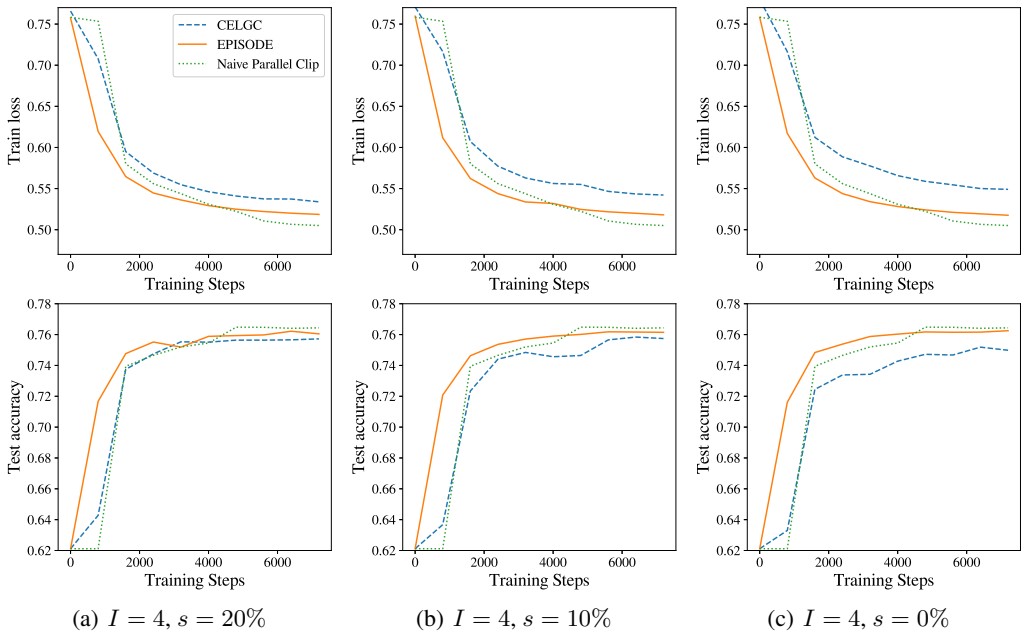

Figure 8: Training curves for all Sentiment140 experiments over training steps.

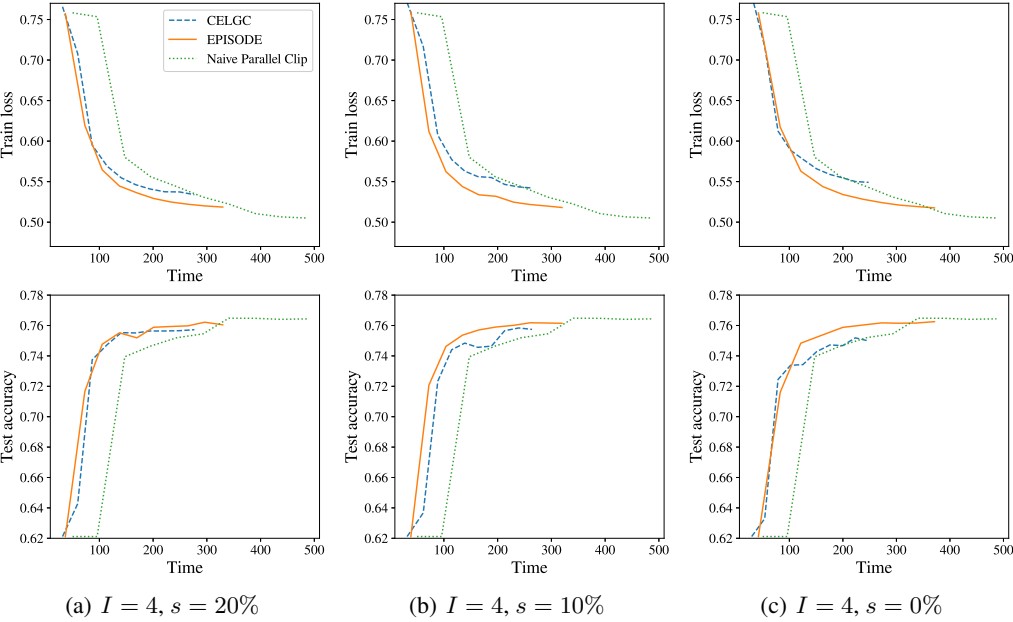

Figure 9: Training curves for all Sentiment140 experiments over running time.

Overall, EPISODE is able to nearly match the training loss and testing accuracy of NaiveParallelClip while requiring significantly *less running time*, and the performance of EPISODE does not degrade as the client data similarity $s$ decreases. Figure 8 shows that, with respect to the number of training steps, EPISODE remains competitive with NaiveParallelClip and outperforms CELGC. In particular, the gap between EPISODE and CELGC grows as the client data similarity decreases, showing that EPISODE can adapt to data heterogeneity. On the other hand, Figure 9 shows that, with a fixed time budget, EPISODE is able to reach lower training loss and higher testing accuracy than both CELGC

and NaiveParallelClip in all settings. This demonstrates the superior performance of EPISODE in practical scenarios.

