# OpenReview forum: "EPISODE: Episodic Gradient Clipping with Periodic Resampled Corrections for Federated Learning with Heterogeneous Data"
_ICLR.cc/2023/Conference — ICLR 2023 poster_

### Official Review · Reviewer_2GmL · 2022-10-22

**Confidence:** 2
**Correctness:** 4
**Technical Novelty And Significance:** 4
**Empirical Novelty And Significance:** 4
**Recommendation:** 8

**Clarity, Quality, Novelty And Reproducibility:**

The paper is well-written and presents the contributions clearly. The results seem novel, and the experiments are well documented.

**Strength And Weaknesses:**

The task studied in the paper is natural and relevant and has been studied in prior works. The paper significantly improves the state-of-the-art results, and thus the contribution is significant. The algorithm is based on novel ideas, and its theoretical analysis is non-trivial. Finally, the experiments demonstrate the effectiveness of the algorithm compared to prior approaches.

I cannot point to weaknesses, although I should mention that I am not familiar enough with the relevant literature, and hence this review has low confidence.


**Summary Of The Paper:**

The paper introduces a new optimization algorithm for the federated learning setting with heterogeneous data, limited communication, and a relaxed smoothness condition. The algorithm uses two new ideas: episodic gradient clipping and periodic resampled corrections. They prove that the algorithm has improved performance for heterogeneous data compared to prior methods, and matches prior results for homogeneous data. They demonstrate the effectiveness of the algorithm empirically, on both synthetic and realistic data.



**Summary Of The Review:**

For the reasons discussed above, I recommend acceptance.

----------------------------------------------------------

Post rebuttal: I have read the other reviews and responses, and I will stick with my original score.

---

> ### Author Response · Authors · 2022-11-18
> **Thank you for taking the time to review our work.**
>
> We are glad that you found our work to be worthy of acceptance. Please let us know if you have any more thoughts or questions.

---

### Official Review · Reviewer_5vyJ · 2022-10-26

**Confidence:** 4
**Correctness:** 3
**Technical Novelty And Significance:** 3
**Empirical Novelty And Significance:** 2
**Recommendation:** 6

**Clarity, Quality, Novelty And Reproducibility:**

The paper is generally well written and easy to follow.

The algorithm seems to be easy to implement for reproducibility, though releasing code if possible could be a bonus.

Clipping has been studied in federated learning for robustness [Learning from History for Byzantine Robust Optimization], and privacy [Understanding clipping for federated learning: Convergence and client-level differential privacy ] with convergence proof. The authors may cosinder weaken the claim on clipping in FL.


**Strength And Weaknesses:**

I did not check the full details of the proof, but the results look reasonable to me. The proposed EPISODE method appears to be novel to me. Compared to CELGC (Liu et al. 2022), EPISODE studies heterogeneity in federated learning, and introduced control variates similar to SCAFFOLD (Karimireddy et al., 2020). Compared to SCAFFOLD, EPISODE studies the assumption of relaxed smoothness functions and applies gradient clipping.

I did not see obvious flaws in the draft. However, I also want to see more discussion connecting theory and practice: in the relaxed smoothness setting, could the authors comment on the theoretical advantage of EPISODE over SCAFFOLD, and how does that connect to the empirical results?

FedAvg is a strong baseline, but I am a little surprised to see that in Figure2, both CELGC and SCAFFOLD underperform FedAvg for accuracy, but EPISODE (the conceptual combination of CELGC and SCAFFOLD) outperforms FedAvg.

I would encourage the authors to discuss application scenarios. Assuming all clients participate in training every round seems to suggest the algorithm is more applicable to cross-silo setting. See [A Field Guide to Federated Optimization https://arxiv.org/abs/2107.06917 Section 3.1].



**Summary Of The Paper:**

This paper proposed the EPISODE method for clipping in federated learning. Each algorithmic round in EPISODE consistents two communication rounds: (1) a “global” gradient is estimated by sampling and averaging the gradient over all clients; (2) the estimated gradient is then used for two purposes, as control variable similar to SCAFFOLD, and as an indicator for gradient clipping; the indicator will determine whether the gradient (calibrated by control variable) for each local step will be clipped. Experiments on SNLI, ImageNet and CIAFR-10 show that EPISODE outperforms FedAvg, SCAFFOLD and CELGC.


**Summary Of The Review:**

As the main contribution seems to be theoretical, I would see more discussion on the theory advantage and the connection between theory and practice.

---

> ### Author Response · Authors · 2022-11-18
> **Thank you for taking the time to review our work and provide helpful comments. See the text below for responses to the points you raised in your review. (cont'd)**
>
> **Q4: “releasing code if possible could be a bonus.”**
>
> A: The code was already included in the original submission as supplementary material. We have updated the supplementary material by providing code for our new experiments (e.g., sentiment140 classification, SNLI running time + ablation).
>
> **Q5: “Clipping has been studied in federated learning for robustness [Learning from History for Byzantine Robust Optimization], and privacy [Understanding clipping for federated learning: Convergence and client-level differential privacy ] with convergence proof. The authors may consider weaken the claim on clipping in FL.”**
>
> A: We would like to emphasize that we only claim that EPISODE is the first algorithm for optimizing non-convex relaxed smooth functions in the heterogeneous FL setting, and not that we are the first to utilize gradient clipping in federated learning. While other work in federated learning has utilized clipping, we are not aware of any works which consider the problem of relaxed smooth functions in heterogeneous FL. We will make it more clear in the revision.
>
> Thank you again for your comments, and please let us know if you have further thoughts.
>
> [1] Zhang, Jingzhao, et al. "Why gradient clipping accelerates training: A theoretical justification for adaptivity." arXiv preprint arXiv:1905.11881 (2019).
>
> [2] Crawshaw, Michael, et al. "Robustness to Unbounded Smoothness of Generalized SignSGD." arXiv preprint arXiv:2208.11195 (2022).

---

> ### Author Response · Authors · 2022-11-18
> **Thank you for taking the time to review our work and provide helpful comments. See the text below for responses to the points you raised in your review.**
>
> **Q1: “However, I also want to see more discussion connecting theory and practice: in the relaxed smoothness setting, could the authors comment on the theoretical advantage of EPISODE over SCAFFOLD, and how does that connect to the empirical results?”**
>
> A: In theory, SCAFFOLD does not have convergence guarantees in the relaxed smoothness setting. The reason is due to quick changes in the gradient. Since the rate of change of the gradient is not bounded under relaxed smoothness, the gradients from the previous communication round (used by SCAFFOLD to compute control variates) may not be accurate estimates of the gradients in the beginning of the current round. See the paragraph titled “Main Challenges” in section 4.1 of the original submission for more discussions. In contrast, we proved that our algorithm EPISODE can converge with linear speedup and reduced communication rounds in the relaxed smoothness setting.
>
> In practice, previous works [1, 2] have provided empirical evidence that training RNNs and Transformers both fall into the relaxed smoothness setting. Further, as noted in Section 5.1 of the original submission, we observed that using SCAFFOLD to train RNNs in our SNLI experiments simply diverged due to a lack of clipping or normalization. In the revised version, we have performed a careful ablation study in Appendix G to show the divergence results.
>
> **Q2: “FedAvg is a strong baseline, but I am a little surprised to see that in Figure2, both CELGC and SCAFFOLD underperform FedAvg for accuracy, but EPISODE (the conceptual combination of CELGC and SCAFFOLD) outperforms FedAvg.”**
>
> A: Thank you for pointing it out. It is indeed true if we take a look at the test accuracy in the setting $I=64$, $s=50\%$, but this is not a general statement for our experiments. The relative performance of FedAvg, CELGC, and SCAFFOLD is highly dependent on the communication interval $I$, the data similarity $s$, and the metric by which the algorithms are compared (e.g., test accuracy), whereas EPISODE is the best performing algorithm in all of our ImageNet settings.
>
> Further, EPISODE is more than just the combination of CELGC+SCAFFOLD: there are two differences between EPISODE and CELGC+SCAFFOLD. The first difference is in the form of the control variates. SCAFFOLD computes control variates by averaging gradients computed across the previous round, whereas EPISODE computes new gradients at the beginning of each round. This means that the control variates in EPISODE are “more recent” than that of SCAFFOLD, potentially providing a better update direction. This aspect of the algorithm design was crucial in our convergence analysis under the relaxed-smoothness setting, so it may play a large part in the empirical performance. In particular, the difference may be exaggerated when the communication interval $I$ is large, since the distance between model parameters across a single round may become larger and larger as the interval grows. In our ImageNet experiments we use $I=64$, which is large relative to our other experiments. The second difference is in the form of clipping. CELGC performs clipping on a step-by-step basis, whereas EPISODE determines clipping on a round-by-round basis. EPISODE’s clipping mechanism is based on global information collected at the beginning of each round, whereas CELGC and CELGC+SCAFFOLD clip only based on local information during the round, which may not be accurate compared to the global average gradient.
>
> We have provided an additional ablation study in Appendix G that compares EPISODE with the direction combination of CELGC+SCAFFOLD for the SNLI dataset, and the results show that EPISODE outperforms CELGC+SCAFFOLD. Please see Appendix G for detailed results.
>
> **Q3: “I would encourage the authors to discuss application scenarios. Assuming all clients participate in training every round seems to suggest the algorithm is more applicable to cross-silo setting.”**
>
> A: Thank you for this suggestion. We have mentioned the applicability to the cross-silo setting in the conclusion of the main text (i.e., Section 6). We have also included some additional cross-silo experiments on the Sentiment140 dataset of the LEAF benchmark in Appendix H, where we show that EPISODE consistently outperforms other baselines.

---

### Official Review · Reviewer_njR7 · 2022-10-31

**Confidence:** 3
**Correctness:** 3
**Technical Novelty And Significance:** 2
**Empirical Novelty And Significance:** 2
**Recommendation:** 5

**Clarity, Quality, Novelty And Reproducibility:**

The paper is well written, theoretical result is novel. However experimental evaluation is limited.

**Strength And Weaknesses:**

+ Paper provides the algorithm that can achieve theoretically faster convergence than previous algorithms under (L_0, L_1)-smoothness.

- It is not compared in the paper how does proved convergence rate compare to FedAvg and to Scaffold convergence rates.

Experimental evaluation is limited:

- IMAGENET and SNLI datasets are not Federated Learning datasets. Would be more interesting to see comparison for some classical FL datasets, e.g. from LEAF benchmark.

- in Imagenet experiments the learning rate was fixed to be the same value for all of the algorithms: this might be not a fair comparison of the algorithms, as optimal stepsizes might be different for different algorithms.

- Paper did not provide experimental comparison to other clipping baselines, such as e.g. adaptive clipping [Andrew et al 2021].

- Every round of the proposed algorithm requires communicating gradients twice. Thus it can be considered that every round has effectively two rounds of communications. This was not taken into account during experimental comparison.

- It is also unclear if the experimental speedup comes from clipping, or from better estimating correction terms G_r and G_r^i.

**Summary Of The Paper:**

The paper proposes a new algorithm called EPISODE for Federated learning with clipping. The algorithm at every round first computes the full gradient from all the clients and after that either clips all of the local updates, or doesn't clip it based on the norm of this full gradient.

Authors prove the convergence under (L_0, L_1) smoothness assumption. Authors also provide an experimental comparison of their algorithm to the previous baselines.

**Summary Of The Review:**

The paper is well written, theoretical result is novel, but not well compared to the literature. Experimental evaluation is limited.

---

> ### Author Response · Authors · 2022-11-18
> **Thank you for taking the time to review our paper and provide insightful comments. We have replied to each of your thoughts below. (cont'd)**
>
> **Q5: “Every round of the proposed algorithm requires communicating gradients twice. Thus it can be considered that every round has effectively two rounds of communications. This was not taken into account during experimental comparison.”**
>
> A: Thank you for pointing this out. We acknowledge that EPISODE indeed requires two communication operations at the beginning of each communication round. However, this additional cost does not affect the order of computation and communication complexity in theory, and we have provided additional experimental results which show that EPISODE outperforms the baselines with respect to running time (not just number of rounds) in Appendix F.
>
> On the performance in practice: Even with an additional communication operation, EPISODE outperforms NaiveParallelClip and CELGC in terms of running time on the SNLI dataset. We have added these additional experimental results on the SNLI dataset in Appendix F. The Figure 6 and Table 2 show that EPISODE requires less time than both CELGC and NaiveParallelClip to reach high levels of test accuracy on the SNLI dataset, especially when client data heterogeneity is high. Please see Appendix F for a detailed discussion about the results.
>
> On the performance in theory: The communication complexity of EPISODE due to this additional communication operation is simply doubled, and this multiplication by a constant does not change the complexity asymptotically (i.e., when $\epsilon$ is small).
> In summary, EPISODE outperforms baselines, both theoretically and empirically, even with the additional communication operation of EPISODE at each round.
>
> **Q6: “It is also unclear if the experimental speedup comes from clipping, or from better estimating correction terms G_r and G_r^i.”**
>
> A: The experimental speedup comes from both clipping and better estimating correction terms. We have provided additional experimental results in Appendix G addressing this concern.
>
> We ran an ablation study on the two components of EPISODE (resampled corrections and episodic clipping) by removing each of these components one at a time and evaluating the resulting algorithm on SNLI. To judge the effect of better correction terms, we evaluated SCAFFOLD with gradient clipping, and to judge the effect of clipping, we evaluated EPISODE with no clipping. The results show that EPISODE with no clipping simply diverges, which shows that clipping is a crucial part of the EPISODE algorithm. On the other hand, SCAFFOLD with clipping does converge, but it consistently reaches a higher training loss and lower testing accuracy than EPISODE, which implies that better estimation of the correction terms is an important part of EPISODE’s success. Please see Appendix G for more discussion of these results.
>
> Thank you again for your time, and please let us know if you have more thoughts.
>
> [1] Yu, Hao, Rong Jin, and Sen Yang. "On the linear speedup analysis of communication efficient momentum SGD for distributed non-convex optimization." International Conference on Machine Learning. PMLR, 2019
>
> [2] Goyal, Priya, et al. "Accurate, large minibatch sgd: Training imagenet in 1 hour." arXiv preprint arXiv:1706.02677 (2017).
>
> [3] Karimireddy, Sai Praneeth, et al. "Scaffold: Stochastic controlled averaging for federated learning." International Conference on Machine Learning. PMLR, 2020.
>
> [4] Zhang, Bohang, et al. "Improved analysis of clipping algorithms for non-convex optimization." Advances in Neural Information Processing Systems 33 (2020): 15511-15521.

---

> ### Author Response · Authors · 2022-11-18
> **Thank you for taking the time to review our paper and provide insightful comments. We have replied to each of your thoughts below.**
>
> **Q1: “It is not compared in the paper how does proved convergence rate compare to FedAvg and to Scaffold convergence rates.”**
>
> A: Thank you for pointing it out. We first want to emphasize that FedAvg and SCAFFOLD might not converge in the relaxed smoothness and heterogeneous setting (We showed that empirically in SNLI experiment, in Appendix G).In addition, our algorithm also converges in the settings where FedAvg and SCAFFOLD converge (e.g., when the function is smooth, $L_0>0$ and $L_1=0$).
>
> We have revised Table 1 in order to explicitly point out the differences in convergence rates in different settings. In the original version, Table 1 contained only the communication complexity ($R$) and the maximum communication complexity $I$ that can ensure linear speedup. The iteration complexity of each algorithm can be computed by $T = RI$. Based on your feedback, we have decided to explicitly include the iteration complexity to clarify the presentation.
>
> From this updated table, we can see that the best iteration complexity of EPISODE (in the heterogeneous case) matches that of the baseline FedAvg (a.k.a, Local SGD) in the smooth case (e.g., $L_0>0$ and $L_1=0$). Similarly, the best iteration complexity of EPISODE (in the relaxed smoothness setting) matches that of SCAFFOLD in the smooth setting. Finally, we can see that EPISODE attains the desirable linear speedup property [1,3], where the iteration complexity is divided by $N$ (where $N$ is the number of clients), so that the convergence rate of EPISODE improves significantly as the number of clients increases. All of this information is contained in Table 1 and discussed in section 4.2. We hope that this clarifies the relative convergence speed of EPISODE with baselines.
>
> **Q2: “IMAGENET and SNLI datasets are not Federated Learning datasets. Would be more interesting to see comparison for some classical FL datasets, e.g. from LEAF benchmark.”**
>
> A: We agree that it would be good to include some standard FL datasets to benchmark the performance of EPISODE. We have included additional experiments on the Sentiment140 dataset from the LEAF benchmark in Appendix H, where we show that EPISODE outperforms other baselines. Please see Appendix H for detailed results.
>
> **Q3: “in Imagenet experiments the learning rate was fixed to be the same value for all of the algorithms: this might be not a fair comparison of the algorithms, as optimal stepsizes might be different for different algorithms.”**
>
> A: Due to computational constraints, we cannot afford to perform extensive hyperparameter tuning for each algorithm on a dataset as large as ImageNet. Instead, we have followed the hyperparameter settings from the well-known and accepted paper [2], which themselves performed extensive tuning to reach their final experimental setup. Previous work [4] on gradient clipping also follows exactly the setup of [2] for imagenet experiment.
>
> **Q4: “Paper did not provide experimental comparison to other clipping baselines, such as e.g. adaptive clipping [Andrew et al 2021].”**
>
> A: The work mentioned on adaptive clipping is a method for differential privacy in federated learning, and our work focuses on nonconvex optimization in federated learning. We agree this is an important work (we have cited this in the first version of our paper), but we believe this work is orthogonal to ours: the purpose of their adaptive clipping mechanism is to achieve differential privacy, and our gradient clipping scheme is to enable efficient optimization for nonconvex and relaxed smooth functions. Therefore we respectfully disagree that we should compare against this baseline. We have compared EPISODE against FedAvg, SCAFFOLD, CELGC, and NaiveParallelClip, which are the most relevant algorithms (clipping or otherwise) to the federated learning problem from the optimization perspective.

---

> ### Author Response · Authors · 2022-12-06
> **Looking Forward to Feedback**
>
> Dear Reviewer njR7,
>
>
> Thank you for your insightful comments that helped us improve our paper. We have addressed all of the questions you specified in your review. We have revised our paper to include: an explicit statement of convergence rate compared to baselines, experiments on the Sentiment140 dataset from the LEAF benchmark, comparison of running time for all algorithms, and an ablation study to analyze the role of each component of the EPISODE algorithm.
>
> Given the deadline of the discussion phase is approaching, if you have any more thoughts or questions, please let us know how we can further address them. If we have answered the concerns that you have, we politely ask that you consider raising the rating of our work. Thank you for your time.
>
> Best,
> Authors

---

### Official Review · Reviewer_m9yX · 2022-10-31

**Confidence:** 4
**Clarity, Quality, Novelty And Reproducibility:** 1. The paper is well written and the …
**Correctness:** 3
**Technical Novelty And Significance:** 3
**Empirical Novelty And Significance:** 2
**Recommendation:** 6

**Strength And Weaknesses:**

Strength:

1. This work seems to be the first to study the federated learning under heterogeneous data setting with relaxed smoothness assumption, and provide a provable algorithm to solve the problem.
2. The paper is well motivated by giving examples why existing algorithms (CELGC and SCAFFOLD) do not work in the setting.
3. The theoretical results come with well sounded implications, making the results convincing (theoretically).

Weaknesses:

1. I am not sure if the given setting is realistic. Like many other distributed optimization papers, this paper put quite a few constraints to the scenarios, such as heterogeneity of the data. But in reality, do we really need to solve such a problem?
2.  While the paper proposes a distributed algorithm, which implies the problem scale is large enough. However, in the experiments, the data/model scale is not that big that a single machine can easily hold them. It is like motivating the paper by solving a very difficult problem but it turns out the problem is not that challenging.

**Summary Of The Paper:**

This paper proposes a communication-efficient distributed gradient clipping algorithm for federated learning, which is called EPISODE. The algorithm works particularly well with the heterogenous data and under the nonconvex and relaxed smoothness setting. The novelty consists in two techniques: episodic gradient clipping and periodic resampled corrections. Another contribution of this paper is to provide a convergence proof for the proposed algorithm under several assumptions. Empirical studies show that EPISODE outperforms other federated learning algorithms on the both synthetic and real datasets, which include SNLI and ImageNet.

**Summary Of The Review:**

The major contribution of this paper is to propose an algorithm to solve the distributed optimization with heterogenous data. Although the setting might not be realistic, the theoretical contribution should still be appreciated.

---

> ### Author Response · Authors · 2022-11-18
> **Thank you for taking the time to review our paper. Below we have addressed each of your concerns.**
>
> **Q1: “I am not sure if the given setting is realistic. Like many other distributed optimization papers, this paper put quite a few constraints to the scenarios, such as heterogeneity of the data. But in reality, do we really need to solve such a problem?”**
>
> A: The main challenges on our problem are heterogeneous data and relaxed smooth loss landscape, and we believe that both of these constraints are of practical and theoretical importance.
>
> There are many examples of practical problems in federated learning with heterogeneous data and relaxed smooth objectives, such as next word prediction [7] and search query suggestion [8] with RNNs.
>
> There are a large number of works in the literature [1, 2, 3] devoted towards the problem of heterogeneity in federated learning due to the prominence of heterogeneous data in practical settings [4], which demonstrates that the issue of heterogeneity in federated learning is accepted in the research community as an important topic of research. As for relaxed smoothness, the importance comes from the need to bridge the gap between theory and practice. A huge number of theoretical works that analyze machine learning problems through the lens of optimization assume the smoothness condition, and this condition is key in proving convergence of various optimization algorithms. However, the empirical evidence provided in [5, 6] shows that some practical neural networks (RNNs, LSTMs, and transformers) do not satisfy the conventional smooth condition, implying that a large number of work on optimization does not apply to these neural networks. By analyzing optimization algorithms under the relaxed smoothness condition, we are providing theoretical analysis that actually applies to these neural networks, thereby bridging the gap between theory and practice.
>
> **Q2: “While the paper proposes a distributed algorithm, which implies the problem scale is large enough. However, in the experiments, the data/model scale is not that big that a single machine can easily hold them. It is like motivating the paper by solving a very difficult problem but it turns out the problem is not that challenging.”**
>
> A: It is true that a single (sufficiently large) machine could hypothetically be used to train on the datasets from our experiments, however, such a setting would violate the principles of federated learning and suffer from decreased performance. Specifically, the federated learning setting is based on the requirement that each user’s data must be kept private and must not leave their device. Therefore, aggregating all user data onto a single machine would violate the constraints of the federated problem, which is not acceptable in practical scenarios.
> Furthermore, even if one decided to aggregate all of the data on a single machine, the training process would lose the runtime advantage gained from parallelization (e.g. linear speedup [3]), and the training process would be significantly slower than if multiple workers had been used. For example, the well-known work [9] uses a distributed setup to train on ImageNet very quickly, and we have used the same scale in our experiment.
> For these two reasons, we believe that the distributed experiments described in our paper (even if they may “fit” onto a single machine) are still well-motivated and important.
> In addition, we have included a new experiment in Appendix H (i.e., sentiment140 data classification), which is included in LEAF (https://arxiv.org/pdf/1812.01097.pdf), a standard federated learning benchmark. We show that our algorithm EPISODE is consistently better than other baselines.
>
> Please let us know if you have any more questions about our paper. Thank you.
>
> [1] Karimireddy, Sai Praneeth, et al. "Scaffold: Stochastic controlled averaging for federated learning." International Conference on Machine Learning. PMLR, 2020.
>
> [2] Sahu, Anit Kumar, et al. "On the convergence of federated optimization in heterogeneous networks." arXiv preprint arXiv:1812.06127 3 (2018): 3.
>
> [3] Yu, Hao, Rong Jin, and Sen Yang. "On the linear speedup analysis of communication efficient momentum SGD for distributed non-convex optimization." International Conference on Machine Learning. PMLR, 2019.
>
> [4] Kairouz, Peter, et al. "Advances and open problems in federated learning." Foundations and Trends in Machine Learning 14.1–2 (2021): 1-210.
>
> [5] Zhang, Jingzhao, et al. "Why gradient clipping accelerates training: A theoretical justification for adaptivity." ICLR 2020.
>
> [6] Crawshaw, Michael, et al. "Robustness to Unbounded Smoothness of Generalized SignSGD." NeurIPS 2022.
>
> [7] Hard, Andrew, et al. "Federated learning for mobile keyboard prediction." arXiv preprint arXiv:1811.03604 (2018).
>
> [8] Yang, Timothy, et al. "Applied federated learning: Improving google keyboard query suggestions." arXiv preprint arXiv:1812.02903 (2018).
>
> [9] Goyal, Priya, et al. "Accurate, large minibatch sgd: Training imagenet in 1 hour." arXiv preprint arXiv:1706.02677 (2017).

---

> > ### Comment · Reviewer_m9yX · 2022-12-01
> > **Thanks for your response.**
> >
> > I have read the response and I am satisfied with it. So I will stick to my original rating.

---

### Author Response · Authors · 2022-11-18
**General Response**

Thank you to all of the reviewers for taking the time to review our paper and help us to improve it. We have responded to each of your reviews individually, and here we provide a general summary of the changes to our paper in response to the reviews. We have marked any changes to the paper in red.

1. In Appendix F, we added a runtime analysis for EPISODE and baselines on the SNLI dataset, showing that EPISODE is superior to all baselines in terms of running time.

2. In Appendix G, we added an ablation study in which we analyze the role of the two key components of the EPISODE algorithm (periodic resampled corrections and episodic clipping), and we concluded that both components are important for the success of EPISODE. In particular, EPISODE achieves good performance while SCAFFOLD and FedAvg fail to converge.

3. In Appendix H, we added an additional experiment on the Sentiment140 dataset of the LEAF benchmark, which is a standard benchmark for federated learning. These results demonstrate the utility of EPISODE in practice.

---

### Decision · Program_Chairs · 2023-01-20

**Decision:**

Accept: poster

**Justification For Why Not Higher Score:**

Some reviewers would have liked to see more discussion on connecting theory and practice.

**Justification For Why Not Lower Score:**

No major concerns remained

**Metareview: Summary, Strengths And Weaknesses:**

This paper proposes a federated learning with clipping of gradients, in the realistic heterogeneous data setting. The algorithm at every round first computes the full gradient from all the clients and depending on the norm of the full gradient decides to clip all local updates. Convergence is shown under the (L_0, L_1) smoothness assumption. Communication is thus doubled compared to the naive FedAvg baseline.

Reviewers liked the clear presentation and ideas, and mostly found the results convincing, both in theory and experiments. Some reviewers would have liked to see more discussion on connecting theory and practice.

We hope the authors will incorporate the several points mentioned by the reviewers in the final version.

**Note From Pc:**

if the above contains the word "oral" or "spotlight" please see: "oral" presentation means -> notable-top-5% and "spotlight" means -> notable-top-25%. As stated in our emails, we are disassociating presentation type from AC recommendations